# Mitigating Modality Prior-Induced Hallucinations in Multimodal Large Language Models via Deciphering Attention Causality

**Guanyu Zhou[1]   Yibo Yan[1,2]   Xin Zou[1]   Kun Wang[3]   Aiwei Liu[1,4]   Xuming Hu[1,2,*]**

[1]The Hong Kong University of Science and Technology (Guangzhou)
[2]The Hong Kong University of Science and Technology
[3]Nanyang Technological University, [4]Tsinghua University
guanyuzhou.ai@gmail.com, xuminghu@hkust-gz.edu.cn

## Abstract

Multimodal Large Language Models (MLLMs) have emerged as a central focus in both industry and academia, but often suffer from biases introduced by visual and language priors, which can lead to multimodal hallucination. These biases arise from the visual encoder and the Large Language Model (LLM) backbone, affecting the attention mechanism responsible for aligning multimodal inputs. Existing decoding-based mitigation methods focus on statistical correlations and *overlook the causal relationships between attention mechanisms and model output*, limiting their effectiveness in addressing these biases. To tackle this issue, we propose a causal inference framework termed **CausalMM** that applies structural **causal** modeling to **MLLM**s, treating modality priors as a confounder between attention mechanisms and output. Specifically, by employing back-door adjustment and counterfactual reasoning at both the visual and language attention levels, our method mitigates the negative effects of modality priors and enhances the alignment of MLLM's inputs and outputs, with a maximum score improvement of **65.3%** on 6 VLind-Bench indicators and **164** points on MME Benchmark compared to conventional methods. Extensive experiments validate the effectiveness of our approach while being a plug-and-play solution. Our code is available at: https://github.com/The-Martyr/CausalMM.

## 1 Introduction

Recent research on Multimodal Large Language Models (MLLMs) has achieved great progress in diverse applications (Yin et al., 2023; Jin et al., 2024; Yan et al., 2024; Zou et al., 2024b), particularly due to their reliance on Transformer models (Vaswani, 2017), where performance is driven by the attention mechanism (Hassanin et al., 2024). In particular, such a mechanism enables the model to assign weights to input information, such as images and text, guiding the generation of outputs. However, the inherent bias in the initial parameters of the model, namely the **modality priors**, can negatively impact output quality via the attention mechanism (Tong et al., 2024a; Zhao et al., 2024; Lee et al., 2024; Chen et al., 2024). In widely used MLLM architectures, attention that most significantly influences output can be divided into two components: visual en-

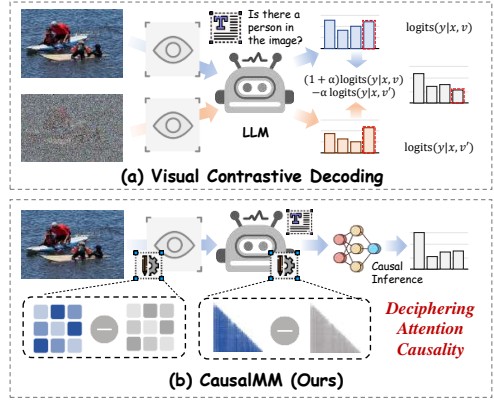

**Figure 1:** The comparison of conventional hallucination mitigation paradigm (*e.g.*, VCD) and our proposed CausalMM.

---

*Corresponding author.

coder attention and Large Language Model (LLM) backbone attention (Liu et al., 2024b). The parametric knowledge of the visual encoder (*i.e.*, **visual priors**) affects the alignment of multimodal information by affecting the visual encoder's attention (Tong et al., 2024a;b). Similarly, the knowledge embedded in the LLM's parameters, referred to as **language priors**, may compromise the model's fidelity to multimodal inputs through attention (Lee et al., 2024). These biases, stemming from the visual encoder and the MLLM's over-reliance on language priors, may lead to issues such as multimodal hallucinations, ultimately degrading model performance (Yang et al., 2023). Several approaches have been proposed to enhance model output without modifying the model weights (Leng et al., 2024; Huang et al., 2024; Zou et al., 2024a). However, as illustrated in Figure 1 (a), existing decoding strategies primarily rely on statistical correlations and predetermined conclusions from posterior analysis to optimize outputs, *without systematically studying the causal relationship between visual attention, language attention, modality priors, and model output*. In this context, the attention mechanism adjusts weights solely based on parameter knowledge, which limits the model's ability to comprehend underlying dependencies in the reasoning process, exacerbates bias, leading to problems such as multimodal hallucinations.

Modality priors are one of the confounding factors in the causal path of MLLM. We introduce a causal reasoning framework CAUSALMM, which can help us better capture the causal impact of effective attention on MLLM output in the presence of these confounding factors, thereby improving the performance of multimodal tasks, as shown in Figure 1 (b). Specifically, we construct a structural causal model (Pearl, 2009) for MLLM, and use intervention and counterfactual reasoning methods under the back-door adjustment paradigm to derive the causal effects of visual and language attention on the model output despite the confounding effect of modal priors. The CAUSALMM method is based on counterfactual reasoning at the visual and language attention levels, which ensures that the model output is more consistent with the multimodal input, thereby mitigating the negative impact of modal priors on performance. Experimental results show that CAUSALMM significantly reduces modal prior bias and improves performance on different tasks, improving **143.7** points on 6 indicators of VLind-Bench, **164** points on the MME Benchmark, and an average improvement of **5.37%** on the three benchmarks of POPE.

Our key contributions can be summarized as follows: ❶ We have constructed a structural causal framework called CAUSALMM flexible for any MLLM, exploring the issues of visual and language priors within the framework. ❷ We apply counterfactual reasoning at the levels of visual and language attention, making the output more aligned with multimodal inputs. ❸ Through comprehensive experiments, we have demonstrated the superior performance of our method in alleviating MLLM hallucinations. In addition, our framework is plug-and-play, and can be integrated with other training-free methods for further improvement.

## 2 RELATED WORKS

**Multimodal Large Language Models.** In recent years, MLLMs have seen significant advancements (Yin et al., 2023; Jin et al., 2024; Huo et al., 2024; Yan & Lee, 2024). Notable works include VITA (Fu et al., 2024b), the first open-source MLLM capable of processing video, image, text, and audio, demonstrating robust performance across various benchmarks. Cambrian-1 (Tong et al., 2024a) is a family of MLLMs designed with a vision-centric approach, achieving state-of-the-art performance and providing comprehensive resources for instruction-tuned MLLMs. Additionally, research on training-free reasoning stage improvements, such as VCD (Leng et al., 2024) and OPERA (Huang et al., 2024), has focused on leveraging human experience to enhance model performance without additional training (Li et al., 2023b; Zheng et al., 2024). In this work, we manage to apply causal reasoning (Pearl, 2009) to make the MLLM automatically optimize the output.

**Causal Inference in Multimodal Learning.** The field of causal inference has seen significant advancements (Pearl, 2009; Xu et al., 2020; Cheng et al., 2023; Gong et al., 2022; Fang & Liang, 2024; Wu et al., 2022), particularly in the context of LLMs and vision systems (Zhang et al., 2023a; Rao et al., 2021; Zhou et al., 2024). Researchers have explored the integration of causal reasoning to enhance the interpretability and robustness of these models (Xu et al., 2020; Zou et al., 2023). For instance, LLMs have been shown to generate accurate causal arguments across various tasks, surpassing traditional methods (Kıcıman et al., 2023). A comprehensive survey has highlighted the potential of causal inference frameworks to improve reasoning capacity, fairness, and multimodality in LLMs (Liu et al., 2024c). Additionally, recent work showcased the use of LLM-guided discovery to significantly improve causal ordering accuracy (Vashishtha et al., 2023). Different from previous

attempts, we tend to use causal reasoning to balance the visual priors and language priors of the model output.

**Modality Priors.** Research on modality priors in MLLMs has seen significant advancements (Tong et al., 2024a; Peng et al., 2023; Lukics & Lukács, 2022; Gema et al., 2024). Studies focused on overcoming language priors by integrating visual modules, enhancing the impact of visual content on model outputs. For instance, (Zhao et al., 2022) proposed a method to improve visual content in Visual Question Answering (VQA) tasks, which proved effective across multiple datasets. Additionally, benchmarks like VLind-Bench (Lee et al., 2024) have been developed to measure language priors in MLLMs, revealing a strong reliance on textual patterns. On the other hand, visual priors have been addressed by augmenting off-the-shelf LLMs to support multimodal inputs and outputs through cost-effective training strategies (Zhang et al., 2024).

## 3 METHODOLOGY

In this section, we construct a structural causal model of MLLM and generate different counterfactual attentions through intervention for counterfactual reasoning based on the back-door criterion.

### 3.1 STRUCTURAL CAUSAL MODEL

We construct a structural causal model (SCM) to describe the relationships among various components of a MLLM (Yang et al., 2021; Pawlowski et al., 2020). In particular, our SCM captures the interactions between the visual and language modalities by modeling causal dependencies among input image ($I$), visual attention ($A_i$), visual token embeddings ($T_i$), language token embeddings ($T_t$), language priors ($P_l$), visual priors ($P_v$), MLLM attention ($A_t$), and model output ($O$).

The causal graph is formulated as follows:

- $I \rightarrow A_i$: The image input $I$ influences the visual attention layer $A_i$.
- $I \rightarrow T_i$: The image input $I$ directly affects the visual token embeddings $T_i$.
- $P_v \rightarrow A_i$: Visual priors $P_v$ contribute to the attention in the visual attention module.
- $P_v \rightarrow T_i$: Visual priors $P_v$ also influence the formation of visual token embeddings $T_i$.
- $A_i \rightarrow T_i$: Visual attention $A_i$ impacts the encoding of visual tokens.
- $T_i \rightarrow O$: Visual tokens $T_i$ contribute directly to the model's output.
- $T_t \rightarrow A_t$: Language token embeddings $T_t$ influence the MLLM's attention $A_t$.
- $T_t \rightarrow O$: Language token embeddings $T_t$ directly impact the final output.
- $P_l \rightarrow A_t$: Language priors $P_l$ inform the MLLM's attention mechanism $A_t$.
- $P_l \rightarrow O$: Language priors $P_l$ directly affect the model output $O$.
- $A_t \rightarrow O$: LLM attention $A_t$ shapes the final output $O$.

In this causal graph, both visual priors ($P_v$) and language priors ($P_l$) serve as confounding factors, influencing the attention layers and embedding representations in both modalities. These priors are mixed into the model and can lead to biased outputs. Our goal is to quantify the causal effect of visual attention ($A_i$) and language attention ($A_t$) on the model output ($O$), while accounting for these confounding effects through intervention and counterfactual reasoning.

### 3.2 INTERVENTION ON MULTIMODAL ATTENTIONS

We perform specific interventions on the attention layers of both the visual and language components to investigate their causal effects on the model's output. These interventions modify the attention weights to generate counterfactual outputs, allowing us to isolate the impact of each modality.

For visual attention, we intervene by replacing the original attention map $A_i$ with a counterfactual state $A_i^*$, expressed as $do(A_i = A_i^*)$. The counterfactual state $A_i^*$ can take various forms, such as random attention weights, uniform distributions, reversed scores, or shuffled attention maps (Rao et al., 2021). Each configuration reveals different aspects of how visual attention influences the output, independent of other factors like the image $I$ and visual processing $P_v$.

Similarly, we intervene in the language attention by applying $do(A_t = A_t^*)$, where $A_t^*$ represents alternative attention states that allow us to explore the impact of the language attention module on the final output, free from the influences of $T_t$, $T_i$, and $P_l$.

The counterfactual attention states are specified as follows:

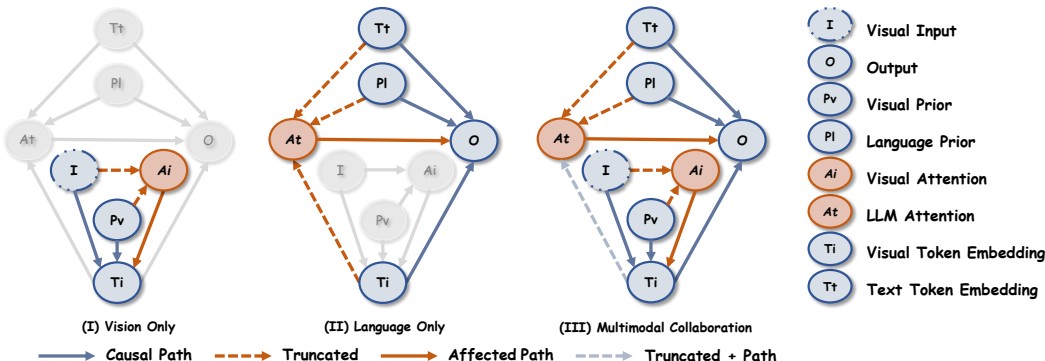

**Figure 2: Causal diagram of counterfactual reasoning.** ❶ In vision-only counterfactual reasoning, we only intervene in visual attention (*i.e.*, the attention of the visual encoder). ❷ In language-only counterfactual reasoning, we only intervene in the multi-head self-attention of LLM. ❸ In multimodal collaborative counterfactual reasoning, we intervene in both visual and language attention at the same time and obtain the sum of their collaborative causal effects.

1. **Random Attention**: Replace the original attention scores with random values drawn from a uniform distribution. For the visual encoder, attention scores $A_i(h, w)$ at spatial locations $(h, w)$ are replaced as follows:

$$A_i'(h, w) = \mathcal{U}(0, 1) \cdot \sigma \cdot \alpha_v, \tag{1}$$

where $\mathcal{U}(0, 1)$ is a random variable drawn from a uniform distribution, $\sigma$ represents the scaling factor for attention, and $\alpha_v$ denotes the normalization parameter. Similarly, for the language model, the random attention values $A_t(n)$ over tokens $n$ are given by:

$$A_t'(n) = \mathcal{U}(0, 1) \cdot \beta \cdot \alpha_l, \tag{2}$$

where $\beta$ is the language attention scaling factor and $\alpha_l$ is the language normalization term.

2. **Uniform Attention**: Assign a constant value to all attention scores. For the visual encoder, the attention at location $(h, w)$ is replaced by the average value:

$$A_i'(h, w) = \frac{1}{H \times W} \sum_{h,w} A_i(h, w) + \epsilon, \tag{3}$$

where $H$ and $W$ represent the height and width of attention map, and $\epsilon$ is a small perturbation added to avoid exact uniformity. For the language model, the attention over $N$ tokens is distributed as:

$$A_t'(n) = \frac{1}{N} \sum_{n=1}^{N} A_t(n) + \delta, \tag{4}$$

where $\delta$ is a small constant ensuring numerical stability.

3. **Reversed Attention**: Invert the attention map by subtracting each attention score from the maximum value of the map. For the visual encoder:

$$A_i'(h, w) = \max(A_i) - A_i(h, w) + \lambda, \tag{5}$$

where $\lambda$ is an offset parameter to control the inversion. For the language model:

$$A_t'(n) = \max(A_t) - A_t(n) + \zeta, \tag{6}$$

where $\zeta$ is the inversion factor for language attention.

4. **Shuffled Attention**: Randomly permute the attention scores across spatial locations for the visual encoder. The new attention map $A_i'$ is created by permuting the original scores $A_i$:

$$A_i'(h, w) = A_i(\pi(h), \pi(w)), \tag{7}$$

where $\pi(h)$ and $\pi(w)$ are random permutations of the height and width indices. This intervention is specific to the visual encoder and does not apply to the language model, as token order is significant in language processing.

By conducting these interventions, we can observe the independent contributions of both visual and language attention to the model's output, controlling for confounding factors such as the image $I$, the tokens $T_t$, and the model's intermediate representations $P_v$ and $P_l$.

### 3.3 COUNTERFACTUAL REASONING

To formalize the impact of counterfactual interventions on the model output, we perform counterfactual reasoning based on the back-door adjustment principle (Pearl, 2009; Li et al., 2023a; Adib et al., 2020; Zhang et al., 2023b). The back-door criterion ensures that we properly account for confounding factors $(I, P_v, P_l)$ when estimating the causal effect of attention mechanisms. Under the framework of back-door adjustment, we are able to effectively obtain the causal effects of other variables under the influence of the confounding factor of modal priors. The specific proof can be found in Sec. A.1. To measure the causal effect of the attention mechanism, we use counterfactual reasoning to simulate the case of attention failure. For the visual attention $(A_i)$:

$$P_{effect\_V} = E_{A_i \sim \tilde{A}_i} \left[ P(O|A_i = \mathbf{A}_i, I = \mathbf{I}, P_v = \mathbf{P}_v) - P(O|\text{do}(A_i = \mathbf{a}_i), I = \mathbf{I}, P_v = \mathbf{P}_v) \right].$$

Here, $P_{effect\_V}$ represents the causal effect of the visual attention mechanism on the model output $O$. The term $\mathbf{A}_i$ denotes the observed visual attention, whereas $\mathbf{a}_i$ represents the intervention applied to the visual attention. For vision-only:

$$t_{next,v} = \arg\max_i \left( \frac{e^{\max(\ell_i + \gamma(\ell_i - \ell_{\text{cf\_v},i}) - \log(\epsilon) - \max_j \ell_j, -\infty)}}{\sum_j e^{\max(\ell_j + \gamma(\ell_j - \ell_{\text{cf\_v},j}) - \log(\epsilon) - \max_k \ell_k, -\infty)}} \right).$$

In this equation, $t_{next,v}$ indicates the index of the next token chosen based solely on visual attention. The variable $\ell_i$ stands for the original logits of the $i$-th token, and $\ell_{\text{cf\_v},i}$ is the counterfactual logit derived from the visual modality. $\gamma$ represents the degree of confidence in the treatment effect. "j" iterates over all tokens in the denominator (to compute the softmax normalization). For the LLM attention $(A_t)$:

$$P_{effect\_L} = E_{A_t \sim \tilde{A}_t} \left[ P(O|A_t = \mathbf{A}_t, T_t = \mathbf{T}_t, P_l = \mathbf{P}_l) - P(O|\text{do}(A_t = \mathbf{a}_t), T_t = \mathbf{T}_t, P_l = \mathbf{P}_l) \right],$$

Where $P_{effect\_L}$ denotes the causal effect of the language model attention on the output $O$. The notation $\mathbf{A}_t$ is the observed language model attention, and $\mathbf{a}_t$ is the intervention applied to the language model attention. For language-only:

$$t_{next,l} = \arg\max_i \left( \frac{e^{\max(\ell_i + \gamma(\ell_i - \ell_{\text{cf\_l},i}) - \log(\epsilon) - \max_j \ell_j, -\infty)}}{\sum_j e^{\max(\ell_j + \gamma(\ell_j - \ell_{\text{cf\_l},j}) - \log(\epsilon) - \max_k \ell_k, -\infty)}} \right).$$

This equation describes the selection of the next token $t_{next,l}$ based purely on language attention. Here, $\ell_i$ is the original logits of the $i$-th token, and $\ell_{\text{cf\_l},i}$ is the counterfactual logit derived from the language modality. In a multimodal setting, the combined causal effect is given by:

$$P_{effect\_M} = E_{A_i, A_t \sim \tilde{A}_i, \tilde{A}_t} \left[ P(O|A_i = \mathbf{A}_i, A_t = \mathbf{A}_t, I = \mathbf{I}, T_t = \mathbf{T}_t, P_v = \mathbf{P}_v, P_l = \mathbf{P}_l) \right]$$
$$- P(O|\text{do}(A_i = \mathbf{a}_i), \text{do}(A_t = \mathbf{a}_t), I = \mathbf{I}, T_t = \mathbf{T}_t, P_v = \mathbf{P}_v, P_l = \mathbf{P}_l),$$

Where $P_{effect\_M}$ represents the combined causal effect of both visual and language attention mechanisms on the output $O$. When integrating visual and language modalities enhanced by counterfactual reasoning, the final token selection is determined by:

$$t_{next} = \arg\max_i \left( \frac{e^{\max(\ell_i + \gamma((\ell_i - \ell_{\text{cf\_v},i}) + (\ell_i - \ell_{\text{cf\_l},i})) - \log(\epsilon) - \max_j \ell_j, -\infty)}}{\sum_j e^{\max(\ell_j + \gamma((\ell_j - \ell_{\text{cf\_v},j}) + (\ell_j - \ell_{\text{cf\_l},j})) - \log(\epsilon) - \max_k \ell_k, -\infty)}} \right).$$

This equation defines the final token selection $t_{next}$ by integrating the effects of both visual and language attention mechanisms, thereby mitigating the negative influence of priors in both modalities and enabling more robust decoding strategies. In all experiments we use direct sampling.

## 4 EXPERIMENTS

In this section, we verify the effectiveness of the CAUSALMM on different benchmarks and implement ablation for different categories of counterfactual attention and number of intervention layers. The case study and gpt-aided-evaluation are in 4.4.

### 4.1 EXPERIMENTAL SETUP

#### 4.1.1 BENCHMARKS

**VLind-Bench.** VLind-Bench (Lee et al., 2024) is a benchmark designed to measure language priors in MLLMs. It disentangles language priors from commonsense knowledge (CK), visual perception (VP), and commonsense biases (CB). There is significant reliance on language priors across models, and the Pipeline Score (SLP) offers insights beyond task-level evaluation.

**POPE.** POPE (Polling-based Object Probing Evaluation) (Li et al., 2023c) is a benchmark for evaluating MLLMs in accurately determining the presence or absence of specific objects in images, assessing object-level hallucination. The framework utilizes Y/N questions derived from object annotations. Evaluation metrics include standard binary classification measures — accuracy, precision, recall, and F1 score — offering a clear quantitative assessment of MLLM performance in distinguishing real from hallucinated objects.

**MME.** MME (Multimodal Large Language Model Evaluation) benchmark (Fu et al., 2024a) quantitatively assesses MLLMs across ten perception-related and four cognition-focused subtasks. To measure object-level hallucination, it uses subsets focused on object existence and count, while attribute-level hallucinations are assessed through subsets concerning object position and color.

### 4.1.2 BASELINES

**Regular setting.** We use two baseline MLLMs LLaVa-1.5 (Li et al., 2023c; Liu et al., 2024a) and Qwen2-VL (Wang et al., 2024) for our baseline setting.

**VCD.** Visual Contrastive Decoding (Leng et al., 2024) is a training-free technique that mitigates object hallucinations in MLLMs. By contrasting output distributions from original and distorted visual inputs, VCD reduces the model's over-reliance on statistical biases and unimodal priors.

**OPERA.** Over-trust Penalty and Retrospection-Allocation (Huang et al., 2024) is an decoding-based method that mitigates hallucinations in MLLMs. It introduces a penalty term during beam search to address over-trust issues, and incorporates a rollback strategy for token selection.

### 4.2 MAIN RESULTS

**Results on VLind-Bench.** As shown in the figure 3, the experimental results on the VLind-Bench benchmark (Lee et al., 2024) are particularly interesting. On the LLaVA-1.5 model, other methods failed to achieve significant performance improvements in balancing modality priors, while the performance under the multimodal collaborative setting has made a significant leap, indicating that the visual priors and language priors of LLaVA-1.5 are balanced. The visual priors of the Qwen2-VL model has been improved, so that the language setting and the multimodal collaborative setting have achieved similar optimal performance.

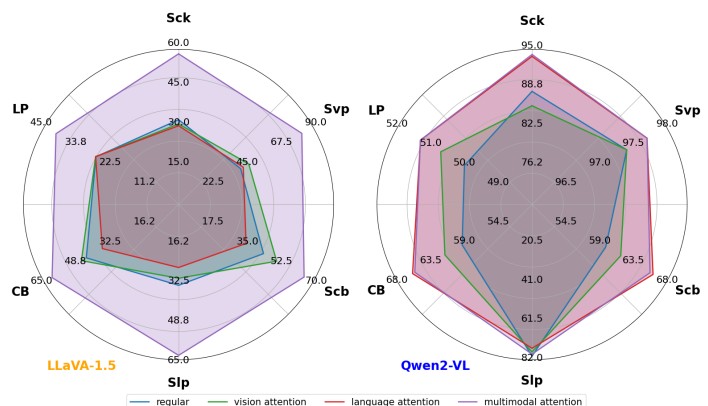

Figure 3: **Scores of different methods on VLind-Bench.** CAUSALMM method significantly improves the model's score on VLind-Bench.

This observation can be attributed to the nature of VLind-Bench, which comprises a suite of evaluation frameworks designed to elucidate the influence of various factors and to quantify the reliance on language priors. Such an evaluation paradigm imposes stringent requirements on the equilibrium of the model's multimodal prior knowledge. Our multimodal collaborative method has notably enhanced the baseline model's performance across all metrics, effectively achieving a balance in the model's modal priors. Compared with other methods that follow human priors, the CAUSALMM method's automatic capture of the causal effect of attention enables it to balance the bias of different modalities simultaneously. This outcome robustly substantiates the efficacy of our methodology (Liu et al., 2024c).

**Results on POPE.** The experimental analysis conducted on the POPE benchmark (see Table 1), as delineated in prior studies (Li et al., 2023c; Lin et al., 2014; Schwenk et al., 2022; Hudson & Manning, 2019), reveals that our proposed CAUSALMM demonstrates superior performance in mitigating object-level hallucinations across random, popular, and adversarial settings. CAUSALMM consistently outperforms existing baselines on the most evaluation metrics, indicating a robust enhancement in performance, with an average metric improvement of **5.37%**.

**Table 1: Main results on POPE tasks.** We evaluate the POPE task accuracy of various MLLMs on the MSCOCO, A-OKVQA, and GQA datasets with LLaVa-1.5 under different decoding settings. **Regular** refers to the scenario where direct sampling is applied. **Vision**, **Language** and **Multimodal** refer to vision-only, language-only, and multimodal collaboration variants of CAUSALMM. The **bold** and the underlined refer to the highest and second highest metrics under each setting, respectively. Each value is followed by the difference relative to regular setting.

| Dataset | Setting | Method | Accuracy | Precision | Recall | F1 Score |
|---|---|---|---|---|---|---|
| MSCOCO | Random | Regular | 83.53 (0.00) | 92.12 (0.00) | 73.33 (0.00) | 81.66 (0.00) |
| | | VCD | 86.40 (2.87) | 94.68 (2.56) | 77.13 (3.80) | 85.01 (3.35) |
| | | OPERA | **89.20** (5.67) | 92.68 (0.56) | 85.26 (11.9) | **88.81** (7.15) |
| | | **Vision** | 86.46 (2.93) | **96.27** (4.15) | 75.86 (2.53) | 84.86 (3.20) |
| | | **Language** | 88.00 (4.47) | 95.96 (3.84) | 79.33 (6.00) | 86.86 (5.20) |
| | | **Multimodal** | 88.93 (5.40) | 95.20 (3.08) | 82.00 (8.67) | 88.10 (6.44) |
| | Popular | Regular | 81.10 (0.00) | 87.89 (0.00) | 72.13 (0.00) | 79.23 (0.00) |
| | | VCD | 83.53 (2.43) | 89.29 (1.40) | 76.20 (4.07) | 82.23 (3.00) |
| | | OPERA | 86.83 (5.73) | 88.24 (0.35) | 85.26 (13.1) | 86.62 (7.39) |
| | | **Vision** | 84.56 (3.46) | 91.57 (3.68) | 76.13 (3.00) | 83.14 (3.91) |
| | | **Language** | 87.03 (5.93) | **91.80** (3.91) | 88.13 (16.0) | 87.17 (7.94) |
| | | **Multimodal** | **87.13** (6.03) | 86.35 (1.46) | **88.20** (16.0) | **87.26** (8.03) |
| | Adversarial | Regular | 78.63 (0.00) | 82.96 (0.00) | 72.06 (0.00) | 77.13 (0.00) |
| | | VCD | 81.10 (2.47) | 84.47 (1.51) | 76.20 (4.14) | 80.12 (3.99) |
| | | OPERA | 81.13 (2.50) | 78.79 (4.17) | **85.20** (13.1) | 81.87 (4.74) |
| | | **Vision** | 82.20 (3.57) | 86.64 (3.68) | 76.13 (4.07) | 81.05 (3.92) |
| | | **Language** | 81.73 (3.10) | 86.28 (3.32) | 75.46 (3.40) | 80.51 (3.38) |
| | | **Multimodal** | **83.70** (5.07) | **87.69** (4.73) | 78.40 (6.34) | **82.78** (5.65) |
| A-OKVQA | Random | Regular | 84.03 (0.00) | 87.67 (0.00) | 79.20 (0.00) | 83.22 (0.00) |
| | | VCD | 85.90 (1.87) | 88.27 (0.60) | 82.80 (3.60) | 85.44 (2.22) |
| | | OPERA | 88.23 (4.20) | 86.13 (1.54) | **91.13** (11.9) | 84.59 (1.37) |
| | | **Vision** | 87.66 (3.63) | 90.24 (2.57) | 84.46 (5.26) | 87.25 (4.03) |
| | | **Language** | 85.96 (1.93) | 89.75 (2.08) | 81.20 (2.00) | 85.26 (2.04) |
| | | **Multimodal** | **88.93** (4.90) | **91.89** (4.22) | 85.40 (6.20) | **88.52** (5.30) |
| | Popular | Regular | 80.23 (0.00) | 80.87 (0.00) | 79.20 (0.00) | 80.02 (0.00) |
| | | VCD | 81.96 (1.73) | 81.44 (0.57) | 82.80 (3.60) | 82.11 (2.09) |
| | | OPERA | 83.40 (3.17) | 78.92 (2.05) | **91.13** (11.9) | 84.59 (4.57) |
| | | **Vision** | 84.03 (3.80) | 83.74 (2.87) | 84.46 (5.26) | 84.10 (4.08) |
| | | **Language** | **85.96** (5.73) | 89.75 (8.88) | 81.20 (2.00) | **85.26** (5.24) |
| | | **Multimodal** | 85.70 (5.47) | **92.60** (11.7) | 77.60 (1.60) | 84.43 (4.41) |
| | Adversarial | Regular | 74.26 (0.00) | 72.33 (0.00) | 78.60 (0.00) | 75.33 (0.00) |
| | | VCD | 76.10 (1.84) | 72.90 (0.57) | 83.06 (4.46) | 77.65 (2.32) |
| | | OPERA | 73.90 (0.36) | 67.77 (4.56) | **91.13** (12.5) | **84.59** (9.26) |
| | | **Vision** | 76.86 (2.60) | 73.43 (1.10) | 84.20 (5.60) | 78.44 (3.11) |
| | | **Language** | 77.43 (3.17) | **74.98** (2.65) | 82.33 (3.73) | 78.48 (3.15) |
| | | **Multimodal** | **77.86** (3.60) | 74.41 (2.08) | 84.93 (6.33) | 79.32 (3.99) |
| GQA | Random | Regular | 83.60 (0.00) | 87.11 (0.00) | 78.86 (0.00) | 82.78 (0.00) |
| | | VCD | 85.86 (2.26) | 88.21 (1.10) | 82.80 (3.94) | 85.41 (2.63) |
| | | OPERA | 88.50 (5.90) | 85.45 (1.66) | **92.80** (13.9) | **88.90** (6.12) |
| | | **Vision** | 87.40 (3.80) | 90.53 (3.42) | 83.53 (4.67) | 86.89 (4.11) |
| | | **Language** | 86.56 (2.96) | 90.18 (3.07) | 82.06 (3.20) | 85.93 (3.15) |
| | | **Multimodal** | **88.50** (5.90) | **90.81** (3.70) | 85.66 (6.80) | 88.16 (5.38) |
| | Popular | Regular | 77.86 (0.00) | 77.32 (0.00) | 78.86 (0.00) | 78.08 (0.00) |
| | | VCD | 79.06 (1.20) | 77.04 (0.28) | 82.80 (3.94) | 79.82 (1.74) |
| | | OPERA | 79.80 (1.94) | 73.65 (3.67) | **92.80** (13.9) | 82.12 (4.04) |
| | | **Vision** | 80.80 (2.94) | 79.20 (1.88) | 83.53 (4.67) | 81.31 (3.23) |
| | | **Language** | 79.93 (2.07) | 78.70 (1.38) | 82.06 (3.20) | 80.35 (2.27) |
| | | **Multimodal** | **82.36** (4.50) | **80.36** (2.04) | 85.66 (6.80) | **82.92** (4.84) |
| | Adversarial | Regular | 75.16 (0.00) | 73.31 (0.00) | 79.13 (0.00) | 76.61 (0.00) |
| | | VCD | 76.33 (1.17) | 73.23 (0.08) | 83.00 (3.87) | 77.81 (1.20) |
| | | OPERA | 75.00 (0.16) | 68.43 (4.88) | **92.80** (13.6) | 78.77 (2.16) |
| | | **Vision** | 76.80 (1.64) | 73.43 (0.12) | 84.20 (5.07) | 78.44 (1.83) |
| | | **Language** | 76.60 (1.44) | 74.21 (0.90) | 81.53 (2.40) | 77.70 (1.09) |
| | | **Multimodal** | **79.53** (4.37) | **76.49** (3.18) | 85.26 (6.13) | **80.64** (3.03) |

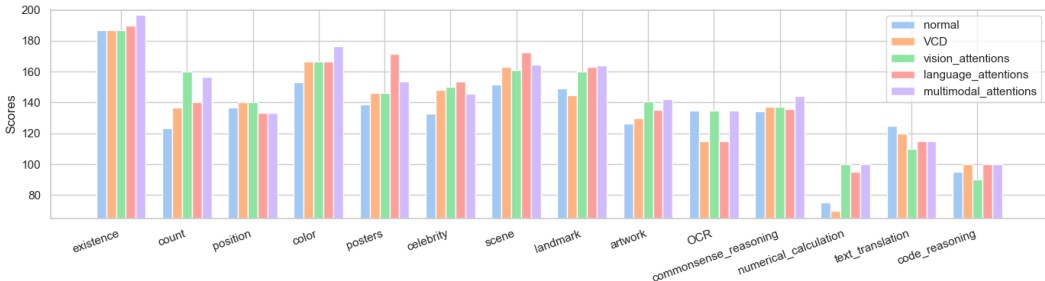

**Figure 4: Result comparison of different categories on MME Benchmark across different methods.** In most tasks, the scores obtained by CAUSALMM are higher than baselines, which verifies its effectiveness.

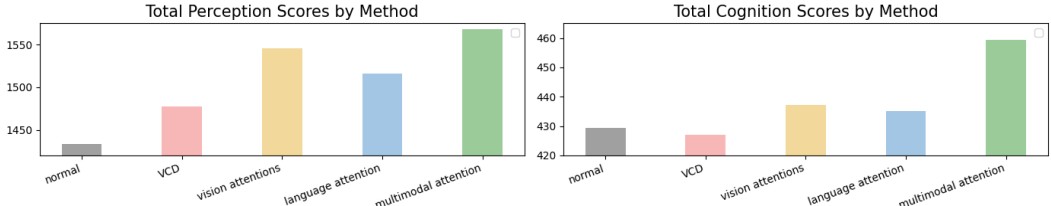

**Figure 5: Result comparison of perception and cognition views on MME Benchmark across different methods.** In both perception and cognition dimensions, variants of CAUSALMM outperform the others.

Notably, both the vision-only and language-only variants of CAUSALMM exhibit significant improvements in effectiveness. Furthermore, the multimodal collaborative approach within our model achieves the highest accuracy, underscoring the synergistic benefits of integrating multiple modalities. Despite the observed performance decline in various baselines when subjected to popular and adversarial settings, our model maintains remarkable stability. This observation suggests that our CAUSALMM method is instrumental in enhancing stability. Moreover, the equilibrium of multimodal parameter priors is deemed crucial, as it can, to a certain extent, amplify the advantages conferred by the balanced priors of distinct modalities. This equilibrium is pivotal in effectively curtailing multimodal hallucinations.

**Results on MME.** The empirical investigations conducted on the MME benchmark (Fu et al., 2024a) offer a thorough assessment of both object-level and attribute-level hallucinations. It has been discerned that while models such as LLaVA-1.5 (Liu et al., 2024b;a) and Qwen2-VL (Wang et al., 2024) exhibit commendable performance in evaluating the presence of objects, they encounter challenges when dealing with more intricate queries, notably those involving counting. As indicated in Figure 4 and Figure 5, our CAUSALMM has been instrumental in significantly enhancing the performance of these models, yielding substantial improvements.

In the domain of attribute-level evaluation, it has been observed that models are more prone to hallucinations concerning attributes like color. Our proposed CAUSALMM, once again, demonstrates significant improvements in this area. The CAUSALMM methods have demonstrated robust performance across various metrics, particularly excelling in numerical computations and counting, which also

**Table 2: Evaluation on the subset of MME perception.** While most of the data are similar, the CAUSALMM method helps Qwen2-VL improve the performance of multiple indicators in MME Benchmark.

| Method | OCR | celebrity | landmark | count |
|---|---|---|---|---|
| Regular | 147.50 | 147.64 | 182.05 | 160.00 |
| **Vision** | 162.50 | 150.29 | 182.75 | 165.00 |
| **Language** | 170.00 | 168.23 | 182.50 | 160.00 |
| **Multimodal** | **170.00** | **168.23** | **182.75** | **165.00** |

translates into an advantage in the overall score. Although the performance on tasks such as Position remains relatively consistent, the overall enhancements in the perception and cognitive categories underscore the effectiveness of these methods in reducing hallucinations.

In the context of poster and scene tasks, the language-only method has achieved the highest performance, which serves as a compelling validation of the impact of language priors on model performance. The MME fullset evaluation corroborates that our CAUSALMM method consistently maintains superior performance across a diverse array of tasks and models, thereby further substantiating its practical utility in enhancing the precision and reliability of MLLMs.

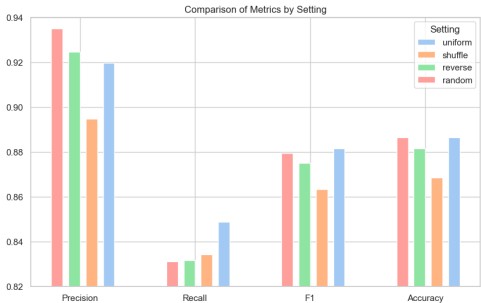

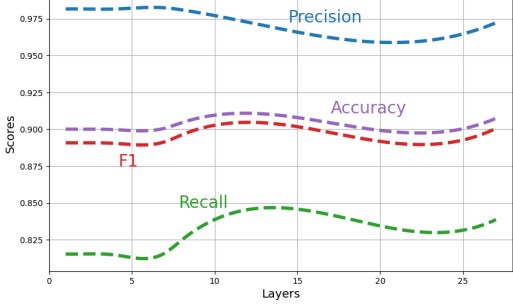

**Figure 6: Ablation on different counterfactual attentions.** The specific value is obtained by taking the average of all the results.

**Figure 7: Ablation on intervention cross layers.** We explored the relationship between the number of layers of intervention in the LLM and the causal effect.

## 4.3 ABLATION STUDY

**Ablation on different counterfactual attention.** To explore the generation of generalized counterfactual attention through interventions (Pearl, 2009), we evaluated four distinct types of counterfactual attention. Ablation experiments were conducted to systematically assess the impact of each type on model performance, as presented in Figure 6. The results demonstrate that using random attention as the anchor for the causal effect leads to the most substantial improvement in model performance. This improvement arises because perturbed attention, when aligned with average attention, can be more clearly distinguished from the original attention. This alignment aligns with the principles of the average causal effect.

The reason for this finding is that perturbed attention, when close to the average attention level, better reflects a generalizable attention distribution pattern. Such generalizability enables a more accurate estimation of the causal effect, as it reduces the influence of outlier attention patterns that may not be representative of the overall dataset. Therefore, this approach more effectively meets the criteria for estimating the average causal effect, contributing to the observed performance improvement.

**Ablation on intervention cross layers.** Beyond the categorization of counterfactuals, the effectiveness of counterfactual attention depends on its application across different layers of a large language model. To investigate the influence of language priors at various depths, interventions were meticulously conducted in the early, middle, and late layers of the model. This multi-layered approach is based on the hypothesis that language priors exert varying levels of influence at different stages of language processing.

By intervening at different layers, we aimed to determine whether counterfactual attention could effectively modulate these priors. Based on the experimental results in Figure 7, interventions between shallow and middle layers proved to be the most effective. We hypothesize that these layers represent the initial stages where language priors significantly impact processing. Interventions in this range can effectively establish anchor points that are influenced by language priors, thereby improving model output to a certain extent.

## 4.4 CASE STUDY

**Case Study on LLaVA-Bench.** To provide a more vivid illustration of the impact of our CAUSALMM method, a case study was conducted on the LLaVA-Bench dataset (Liu et al., 2024b). This study employed specific visual questions and the corresponding model responses to elucidate the enhancement in model output quality and the mitigation of adverse effects, such as hallucinations, attributable to the CAUSALMM method. A representative example is depicted in Figure 8. Objects like *boat*,

**Table 3: GPT-4o-aided-evaluation.** The evaluation results of gpt4-o as an expert. The four indicators represent the overall quality, conversational, detailedness and complexity.

| Method | All | Conv | Detail | Cplx |
|---|---|---|---|---|
| Regular | 84.7 | 87.7 | 89.3 | 80.4 |
| **Vision** | 84.8 | 88.8 | 86.7 | **81.4** |
| **Language** | 84.7 | 88.8 | 88.0 | 80.4 |
| **Multimodal** | **85.0** | **88.8** | **89.3** | 80.0 |

which frequently co-occur with the potential ground truth object *ocean*, are prone to being hallucinated. However, the application of our CAUSALMM method notably diminishes these hallucinatory tendencies. It enables the model to discern the city situated at the base of the volcano while maintaining a coherent and informative output text. This outcome underscores the efficacy of CAUSALMM in refining the output and curtailing the emergence of spurious associations.

**GPT-4o-aided-evaluation.** Supplementing the standard benchmark assessments, we have employed the GPT-4o* as an evaluative referee to quantitatively measure the efficacy of our CAUSALMM method. The evaluation was conducted using a 10-point scoring system, with the results compiled in Table 3. The results indicate that CAUSALMM is more adept at generating responses that align with the sophisticated evaluative standards set by GPT-4o.

---

**POSITIVE CASE**

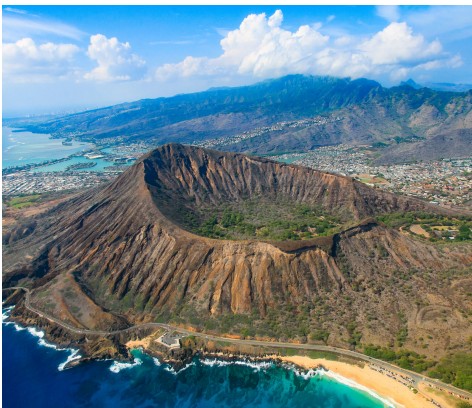

**Regular:** The image showcases a picturesque scene of a mountainous island with a large hill or mountain in the middle. The island is surrounded by a beautiful blue ocean, which adds to the serene atmosphere. The mountain appears to be a volcano, giving it a striking appearance. In the background, there are several small boats scattered across the ocean, likely enjoying the peaceful waters. The combination of the island, mountain, and ocean creates a breathtaking view, making it an ideal location for relaxation and exploration.

**CAUSALMM:** The image features a breathtaking view of a large, rocky island with a lush green hillside. The island is situated in the middle of a body of water, which could be an ocean or a lake. The island's terrain is a mix of rocky cliffs and green vegetation, creating a stunning contrast against the water. In the background, a city can be seen, possibly on the mainland or another island. The city's buildings are visible against the backdrop of the large island, adding to the overall beauty of the scene.

**Figure 8:** Positive case of CausalMM.

---

**NEGATIVE CASE**

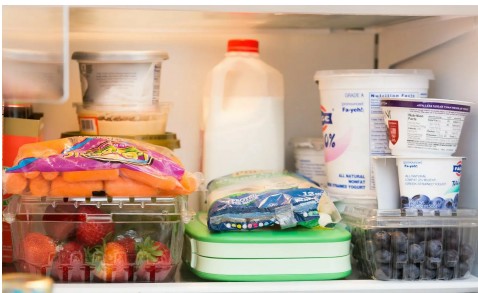

**Prompt:** Is there any strawberry-flavored yogurt in the fridge?

**Regular:** Yes, there is a container of strawberry-flavored yogurt in the fridge.

**CAUSALMM:** Yes, there is a container of strawberry-flavored yogurt in the fridge.

**Figure 9:** Unsolved case of CausalMM.

---

## 5 CONCLUSION

Though promising, MLLMs are prone to biases from visual and language priors, which can degrade performance and cause multimodal hallucinations. These biases stem from the influence of the visual encoder and LLM backbone on the attention mechanism, hindering the model's ability to align multimodal inputs effectively. To overcome this, we introduced a causal reasoning framework termed CAUSALMM that applies structural causal modeling to MLLMs, treating modality priors as a confounding factor. By leveraging back-door adjustment and counterfactual reasoning at both visual and language attention levels, CAUSALMM demonstrates significant reductions in language priors bias and offers a plug-and-play solution compatible with other training-free approaches, providing a insightful path forward for trustyworthy multimodal intelligence.

---

*https://platform.openai.com/docs/models/gpt-4o

## 6 ACKNOWLEDGMENTS

This work was supported by CAAI-Ant Group Research Fund; Guangdong Provincial Department of Education Project (Grant No.2024KQNCX028); Scientific Research Projects for the Higher-educational Institutions (Grant No.2024312096), Education Bureau of Guangzhou Municipality; Guangzhou-HKUST(GZ) Joint Funding Program (Grant No.2025A03J3957), Education Bureau of Guangzhou Municipality.

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

# A APPENDIX

## A.1 FURTHER DEMONSTRATION

### STRUCTURAL CAUSAL MODEL (SCM):

Take the three core variables mentioned in the article as an example.

### VARIABLES AND THEIR ROLES:

- $A$ (attention): This represents the model's attention mechanism that we aim to evaluate or manipulate.
- $M$ (modality priors): Modality priors influence both the model's attention ($A$) and the output ($O$), thus creating confounding.
- $O$ (model output): The outcome variable, which is affected both directly by $A$ and indirectly through $M$.

### CAUSAL STRUCTURE AND BACK-DOOR PATHS:

- The back-door path in this SCM is $A \leftarrow M \rightarrow O$, which starts with an arrow pointing into $A$ and creates a confounding junction structure.
- To isolate the causal effect of $A$ on $O$, the confounding influence of $M$ must be blocked.

### BACK-DOOR CRITERION:

To apply back-door adjustment, the adjustment set $M$ must satisfy the following criteria:

1. $M$ blocks all back-door paths from $A$ to $O$.
2. $M$ does not include any descendants of $A$ (i.e., variables causally influenced by $A$).

By intervening on $A$ and adjusting for $M$, we can isolate the causal effect of $A$ on $O$.

### BACK-DOOR ADJUSTMENT FORMULA:

Given a sufficient adjustment set $M$, the causal effect $P(o \mid do(a))$ is identified as:

$$P(o \mid do(a)) = \sum_m P(o \mid a, m) P(m)$$

### DERIVATION:

1. **Starting with the interventional distribution:**
$$P(o \mid do(a)) = \sum_m P(o \mid do(a), m) P(m \mid do(a))$$

2. **Using the property of the intervention** $do(a)$**:** Under the intervention $do(a)$, the variable $A$ is no longer influenced by $M$. Thus:
$$P(m \mid do(a)) = P(m)$$

3. **Replacing** $P(o \mid do(a), m)$ **with the observational counterpart:** Due to the back-door criterion, $M$ blocks all confounding paths, allowing:
$$P(o \mid do(a), m) = P(o \mid a, m)$$

4. **Combining these results:**
$$P(o \mid do(a)) = \sum_m P(o \mid a, m) P(m)$$

APPLICATION TO ATTENTION-OUTPUT FRAMEWORK:

In the context of our framework:

1. **Back-door path:** The back-door path $A \leftarrow M \rightarrow O$ reflects the confounding effect of modality priors ($M$) on the attention mechanism ($A$) and the model's output ($O$).

2. **Intervention:** By intervening on $A$, we ensure that the causal effect of attention on the output is isolated, free from the influence of modality priors.

3. **Adjustment:** To block the back-door path, we adjust for $M$, computing the summation over all possible values of $M$ to account for its confounding effect.

FULL FORMULA FOR THE FRAMEWORK:

In our framework, the causal effect of attention ($A$) on the model output ($O$) can be computed as:

$$P(o \mid do(a)) = \sum_m P(o \mid a, m)P(m)$$

- $P(o \mid a, m)$: The conditional probability of the output given attention $A$ and modality priors $M$.
- $P(m)$: The marginal probability of modality priors $M$.

By applying the back-door adjustment formula, we mitigate the influence of confounding modality priors, ensuring that the attention mechanism's causal contribution to the output is properly estimated.

## A.2 Additional Experimental Results

To demonstrate the effectiveness of our approach on large multimodal language models of different architectures, we added experimental data from the Q-former-based InstructBLIP model and the embedding-autoregressive-based Chameleon model to the original experimental data from the vision encoder-mlp-llm paradigm. See tab. 4 and tab. 5 for specific data. Comparisons with more baseline methods can be found in tab. 6.

**Table 4: Additional Experimental Results on POPE tasks: Chameleon.** We evaluate the POPE task accuracy of various MLLMs on the MSCOCO, A-OKVQA, and GQA datasets with Chameleon (Team, 2024) under different decoding settings. **Regular** refers to the scenario where direct sampling is applied. **Language** refer to language-only.

| Dataset | Setting | Method | Accuracy | Precision | Recall | F1 Score |
|---------|---------|--------|----------|-----------|--------|----------|
| MSCOCO | Random | Regular | 61.90 | 57.46 | 91.67 | 70.64 |
| | | **Language** | 69.23 | 63.17 | 92.27 | 74.99 |
| | Popular | Regular | 65.10 | 59.86 | 91.67 | 72.43 |
| | | **Language** | 69.43 | 63.34 | 92.27 | 75.12 |
| | Adversarial | Regular | 60.20 | 56.28 | 91.40 | 69.66 |
| | | **Language** | 64.00 | 58.94 | 92.33 | 71.95 |
| A-OKVQA | Random | Regular | 60.37 | 56.26 | 93.20 | 70.16 |
| | | **Language** | 65.70 | 60.14 | 93.13 | 73.08 |
| | Popular | Regular | 57.30 | 54.25 | 93.20 | 68.58 |
| | | **Language** | 63.07 | 58.16 | 93.13 | 71.60 |
| | Adversarial | Regular | 53.57 | 51.99 | 93.20 | 66.75 |
| | | **Language** | 56.83 | 53.96 | 93.13 | 68.33 |
| GQA | Random | Regular | 60.37 | 56.26 | 93.20 | 70.16 |
| | | **Language** | 68.43 | 62.18 | 94.13 | 74.89 |
| | Popular | Regular | 59.37 | 55.76 | 90.67 | 69.05 |
| | | **Language** | 66.73 | 60.81 | 94.13 | 73.89 |
| | Adversarial | Regular | 52.73 | 51.55 | 90.67 | 65.73 |
| | | **Language** | 57.77 | 54.50 | 94.13 | 69.03 |

**Table 5: Additional Experimental Results on POPE tasks: InstructBLIP.** We evaluate the POPE task accuracy of various MLLMs on the MSCOCO, A-OKVQA, and GQA datasets with InstructBLIP (Dai et al., 2023) under different decoding settings. **Regular** refers to the scenario where direct sampling is applied. **Vision**, **Language** and **Multimodal** refer to vision-only, language-only, and multimodal collaboration variants of CAUSALMM.

| Dataset | Setting | Method | Accuracy | Precision | Recall | F1 Score |
|---------|---------|--------|----------|-----------|--------|----------|
| MSCOCO | Random | Regular | 80.71 | 81.67 | 79.19 | 80.41 |
| | | VCD | 84.53 | 88.55 | 79.32 | 83.68 |
| | | **Vision** | 87.17 | 92.72 | 80.67 | 86.27 |
| | | **Language** | 86.90 | 94.89 | 78.00 | 85.62 |
| | | **Multimodal** | 87.90 | 94.59 | 80.40 | 86.92 |
| | Popular | Regular | 78.22 | 77.87 | 78.85 | 78.36 |
| | | VCD | 81.47 | 82.89 | 79.32 | 81.07 |
| | | **Vision** | 83.97 | 86.37 | 80.67 | 83.42 |
| | | **Language** | 83.53 | 87.71 | 78.00 | 82.57 |
| | | **Multimodal** | 84.90 | 88.35 | 80.40 | 84.19 |
| | Adversarial | Regular | 75.84 | 74.30 | 79.03 | 76.59 |
| | | VCD | 79.56 | 79.67 | 79.39 | 79.52 |
| | | **Vision** | 81.47 | 81.89 | 80.80 | 81.34 |
| | | **Language** | 82.00 | 84.73 | 78.07 | 81.26 |
| | | **Multimodal** | 82.43 | 83.71 | 80.53 | 82.09 |
| A-OKVQA | Random | Regular | 80.91 | 77.97 | 86.16 | 81.86 |
| | | VCD | 84.11 | 82.21 | 87.05 | 84.56 |
| | | **Vision** | 87.33 | 85.94 | 89.27 | 87.57 |
| | | **Language** | 87.87 | 87.72 | 88.07 | 87.89 |
| | | **Multimodal** | 88.47 | 87.86 | 89.27 | 88.56 |
| | Popular | Regular | 76.19 | 72.16 | 85.28 | 78.17 |
| | | VCD | 79.78 | 76.00 | 87.05 | 81.15 |
| | | **Vision** | 81.07 | 76.69 | 89.27 | 82.50 |
| | | **Language** | 82.33 | 79.01 | 88.07 | 83.29 |
| | | **Multimodal** | 82.13 | 78.45 | 88.60 | 83.22 |
| | Adversarial | Regular | 70.71 | 65.91 | 85.83 | 75.56 |
| | | VCD | 74.33 | 69.46 | 86.87 | 77.19 |
| | | **Vision** | 74.83 | 69.11 | 89.80 | 78.11 |
| | | **Language** | 76.27 | 71.07 | 88.60 | 78.87 |
| | | **Multimodal** | 75.97 | 70.51 | 89.27 | 78.79 |
| GQA | Random | Regular | 79.65 | 77.14 | 84.29 | 80.56 |
| | | VCD | 83.69 | 81.84 | 86.61 | 84.16 |
| | | **Vision** | 86.10 | 84.56 | 88.33 | 86.40 |
| | | **Language** | 86.67 | 86.86 | 86.40 | 86.63 |
| | | **Multimodal** | 87.23 | 86.67 | 88.00 | 87.33 |
| | Popular | Regular | 73.87 | 69.63 | 84.69 | 76.42 |
| | | VCD | 78.57 | 74.62 | 86.61 | 80.17 |
| | | **Vision** | 77.77 | 72.92 | 88.33 | 79.89 |
| | | **Language** | 79.17 | 75.48 | 86.40 | 80.57 |
| | | **Multimodal** | 78.97 | 74.99 | 86.93 | 80.52 |
| | Adversarial | Regular | 70.56 | 66.12 | 84.33 | 74.12 |
| | | VCD | 75.08 | 70.59 | 85.99 | 77.53 |
| | | **Vision** | 74.50 | 69.33 | 87.87 | 77.51 |
| | | **Language** | 76.30 | 71.81 | 86.60 | 78.51 |
| | | **Multimodal** | 75.83 | 71.19 | 86.80 | 78.22 |

**Table 6: More results on POPE tasks.** We evaluate the POPE task accuracy of various MLLMs on the POPE benchmark with LLaVa-1.5 and InstructBLIP under different decoding settings. In the table, the values taken are the averages of the three parts of the POPE benchmark (MSCOCO, A-OKVQA, GQA). **Regular** refers to the scenario where direct sampling is applied. **Vision**, **Language** and **Multimodal** refer to vision-only, language-only, and multimodal collaboration variants of CAUSALMM. DOLA stands for DoLa: Decoding by Contrasting Layers Improves Factuality in Large Language Models(Chuang et al., 2023).

| Dataset | Setting | Method | Accuracy | Precision | Recall | F1 Score |
|---|---|---|---|---|---|---|
| InstructBLIP | Random | Regular | 80.42 | 78.93 | 83.21 | 80.94 |
| | | DOLA | 83.00 | 83.06 | 83.13 | 83.00 |
| | | VCD | 84.11 | 84.20 | 84.33 | 84.13 |
| | | OPERA | 85.07 | 88.39 | 80.73 | 84.39 |
| | | AGLA | 87.30 | 88.83 | 85.68 | 87.07 |
| | | Vision | 86.87 | 87.74 | 86.09 | 86.75 |
| | | Language | 87.15 | 89.82 | 84.16 | 86.71 |
| | | Multimodal | 87.87 | 89.71 | 85.89 | 87.60 |
| | Popular | Regular | 76.09 | 73.22 | 82.94 | 77.65 |
| | | DOLA | 78.99 | 77.12 | 83.13 | 79.85 |
| | | VCD | 79.94 | 77.84 | 84.33 | 80.80 |
| | | OPERA | 78.33 | 73.85 | 87.73 | 80.20 |
| | | AGLA | 81.86 | 80.17 | 85.68 | 82.58 |
| | | Vision | 80.94 | 78.66 | 86.09 | 81.94 |
| | | Language | 81.68 | 80.73 | 84.16 | 82.14 |
| | | Multimodal | 82.00 | 80.60 | 85.31 | 82.64 |
| | Adversarial | Regular | 72.37 | 68.78 | 83.06 | 75.42 |
| | | DOLA | 74.67 | 71.53 | 83.11 | 76.68 |
| | | VCD | 76.32 | 73.24 | 84.08 | 78.08 |
| | | OPERA | 75.50 | 70.49 | 87.73 | 78.17 |
| | | AGLA | 77.29 | 74.09 | 85.67 | 79.16 |
| | | Vision | 76.93 | 73.44 | 86.16 | 78.99 |
| | | Language | 78.19 | 75.87 | 84.42 | 79.55 |
| | | Multimodal | 78.08 | 75.14 | 85.53 | 79.70 |
| LLaVA-1.5 | Random | Regular | 83.72 | 89.30 | 77.13 | 82.55 |
| | | DOLA | 84.78 | 87.59 | 81.27 | 84.19 |
| | | VCD | 86.05 | 90.39 | 80.91 | 85.29 |
| | | OPERA | 88.64 | 88.09 | 89.73 | 87.43 |
| | | AGLA | 88.54 | 94.41 | 82.08 | 87.71 |
| | | Vision | 87.17 | 92.35 | 81.28 | 86.33 |
| | | Language | 86.84 | 91.96 | 80.86 | 85.68 |
| | | Multimodal | 88.79 | 92.63 | 84.35 | 88.26 |
| | Popular | Regular | 79.73 | 82.03 | 76.73 | 79.11 |
| | | DOLA | 79.75 | 84.11 | 76.22 | 80.61 |
| | | VCD | 81.52 | 82.59 | 80.60 | 81.39 |
| | | OPERA | 83.34 | 80.27 | 89.73 | 84.44 |
| | | AGLA | 85.14 | 87.88 | 82.08 | 84.68 |
| | | Vision | 83.13 | 84.84 | 81.37 | 82.85 |
| | | Language | 84.31 | 86.75 | 83.80 | 84.26 |
| | | Multimodal | 85.06 | 86.44 | 83.82 | 84.87 |
| | Adversarial | Regular | 76.02 | 76.20 | 76.60 | 76.36 |
| | | DOLA | 76.32 | 77.27 | 75.47 | 76.16 |
| | | VCD | 77.84 | 76.87 | 80.75 | 78.53 |
| | | OPERA | 76.68 | 71.66 | 89.71 | 79.46 |
| | | AGLA | 81.13 | 81.20 | 82.10 | 81.36 |
| | | Vision | 78.62 | 77.83 | 81.51 | 79.31 |
| | | Language | 78.59 | 78.49 | 79.77 | 78.90 |
| | | Multimodal | 80.36 | 79.53 | 82.86 | 80.91 |

### A.3 VISUALIZATION OF COUNTERFACTUAL ATTENTIONS

### A.3.1 VISION ATTENTION

In this work, we used four commonly used counterfactual visual attentions (Rao et al., 2021): random, reverse, uniform, and shuffle. They represent taking random values for global attention, reversing global attention, using consistent attention values, and disrupting the original attention distribution. They can all effectively provide anchor points for obtaining causal effects, thereby helping the model improve potential modal priors. Among them, the settings of random and uniform are closest to the average value in value distribution, so they can provide the largest positive average causal effect.

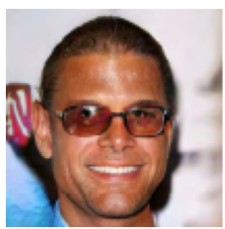
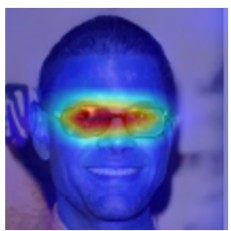

**Figure 10: Normal vision attention of vision encoder.**

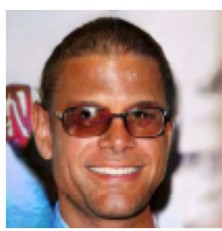
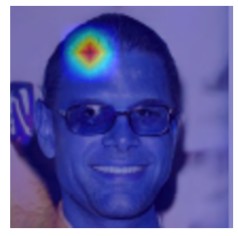
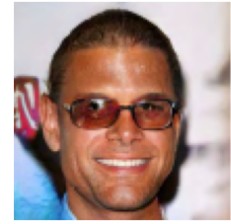
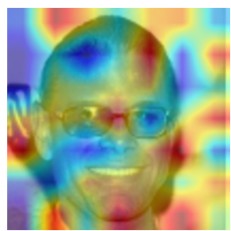

**Figure 11: Shuffled vision attention of vision encoder.**

**Figure 12: Random vision attention of vision encoder.**

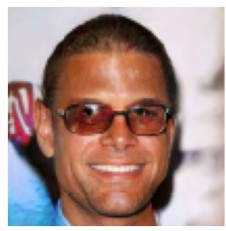
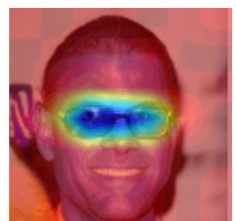
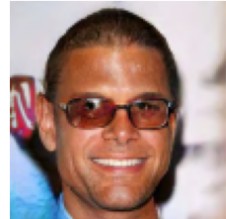
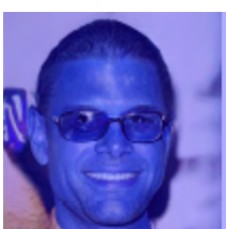

**Figure 13: Reversed vision attention of vision encoder.**

**Figure 14: Uniform vision attention of vision encoder.**

### A.3.2 LANGUAGE ATTENTION

We visualize four similar counterfactual attentions: they represent taking random values for global attention, negating global attention, using consistent attention values, and disrupting the original attention distribution. We take three of them for visualization. Similarly, they can effectively provide anchors for obtaining causal effects, thereby helping the model improve the potential modal prior. Compared with visual attention, large language models with large parameters are not as sensitive to changes in attention as visual encoders.

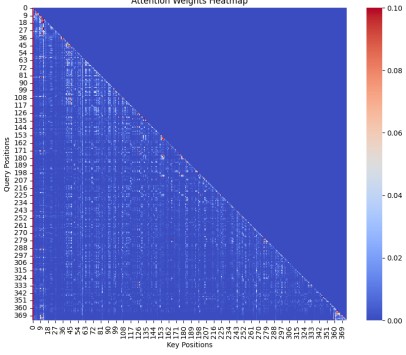

Figure 15: Visualization of normal LLM attention.

Figure 16: Visualization of random LLM attention.

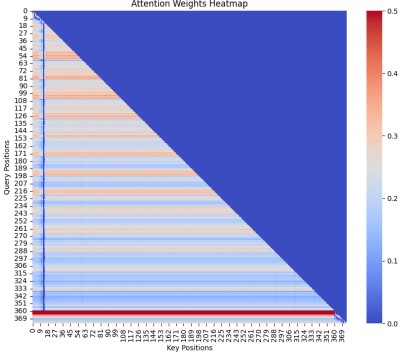

Figure 17: Visualization of reversed LLM attention.

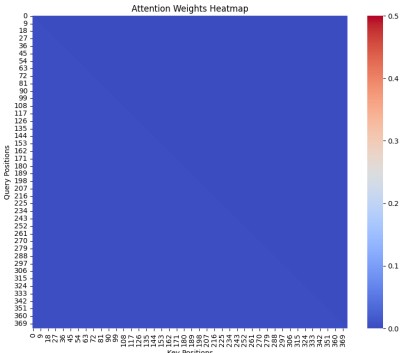

Figure 18: Visualization of uniform LLM attention.

## A.4 CASE STUDY

We have selected some typical cases to demonstrate the effect of our method. The CausalMM method balances different modal priors to weaken the bias that may be caused by the model's own parameter knowledge from the perspective of vision and language, so that the model's output can be more aligned with multimodal input. This improvement is reflected in the model's perception and cognitive ability of specific things, and the potential hallucinations of the original model have been effectively improved.

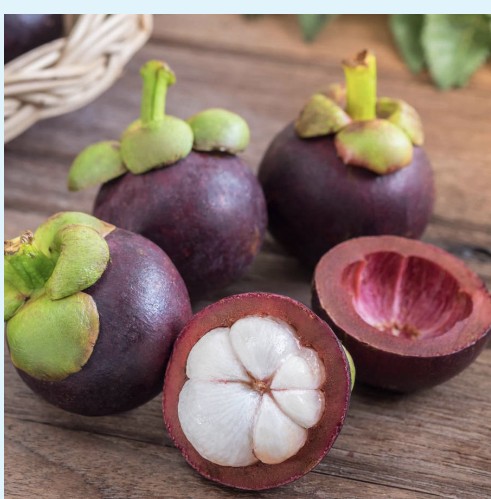

**Prompt:**
How many uncut fruits are in the image?

**Regular:**
There are four uncut fruits in the image.

**Our Method:**
There are three uncut fruits in the image.

Figure 19: Case of counting task.

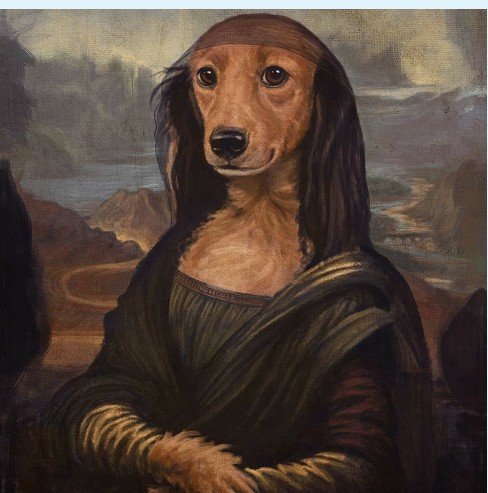

**Prompt:**
Describe this photo in detail.

**Regular:** The image features a painting of a dog wearing a Renaissance-style dress, giving it a human-like appearance. The dog is sitting on a chair, with its head turned to the side, and appears to be looking at the viewer. The painting captures the dog's unique and artistic expression, making it an interesting and eye-catching piece.

**Our Method:** The image features a painting of a dog wearing a Renaissance-style dress and a hat. The dog appears to be sitting down, possibly posing for the portrait. The dog's dress is long, covering its body from the neck down. The painting captures the dog's unique appearance, making it an interesting and creative piece of art.

Figure 20: Case of description task.

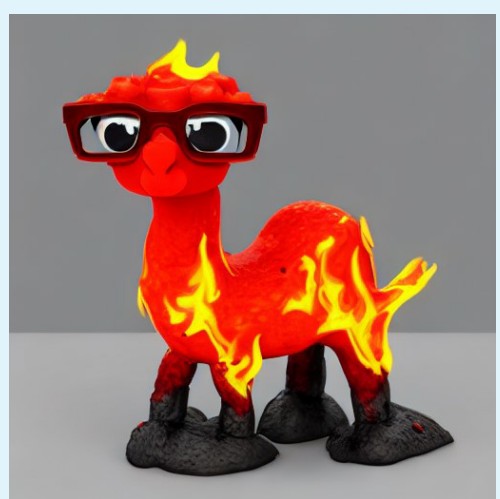

**Prompt:**
What material appears to make up the creature?

**Regular:**
The creature appears to be made of a combination of plastic and glass, as it has a plastic body and glass eyes.

**Our Method:**
The creature appears to be made of plastic.

Figure 21: Case of perception task.

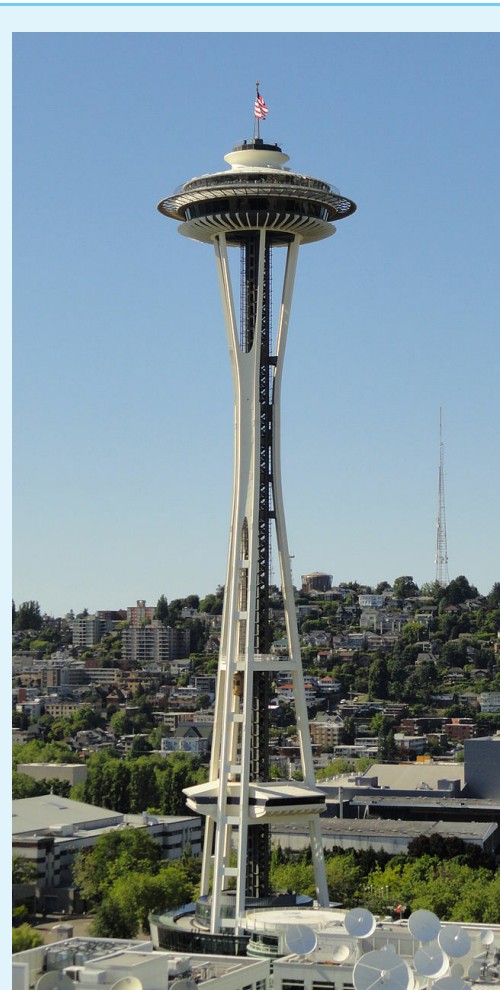

**Prompt:**
Describe this photo in detail.

**Regular:**
The image showcases a tall skyscraper towering over a cityscape. The tower is prominently visible in the middle of the scene, reaching high into the sky. The cityscape around the tower includes numerous buildings of various sizes and shapes, creating a bustling urban atmosphere. There are also several cars scattered throughout the scene, indicating that the area is busy with traffic. Additionally, there are three people visible in the image, likely going about their daily activities or enjoying the view of the towering skyscraper.

**Our Method:**
The image features a tall, skinny building with a large antenna on top of it. The building is located in a city, surrounded by other buildings, and it appears to be a prominent landmark. The antenna is positioned towards the top of the building, making it stand out. In the background, there are several other buildings visible, showcasing the urban landscape. The scene captures the essence of a bustling cityscape with a mix of architectural styles.

Figure 22: Case of description task.

**Limitation of CAUSALMM**

We further evaluated the effect of the CAUSALMM method based on a case study to explore the limitations of the method. The specific example is in fig. 23. We found that even after correcting some of the hallucinations caused by visual and language priors, our method still did not significantly improve the acquisition of high-level semantics. We believe that the bottleneck of our method is the performance bottleneck of the vision encoder and the LLM backbone. In future work, we will explore how to maximize the positive impact of balanced modal priors when the backbone model is fixed.

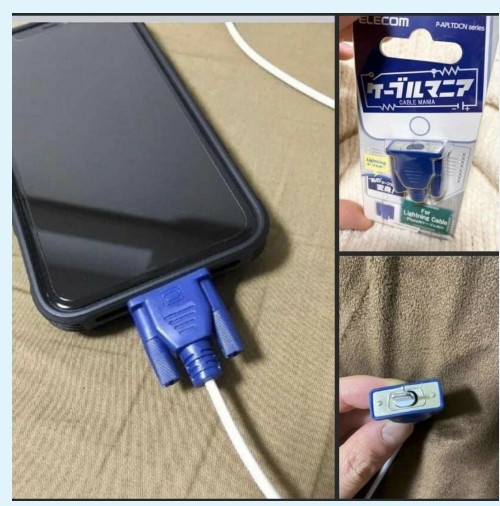

**Prompt:**

What is funny about this image?

Describe it panel by panel.

**Regular:**

The image is a collage of three panels, each showing a different aspect of the same scene. The first panel features a cell phone plugged into a charger, while the second panel shows a person holding a box of earphones. The third panel displays a close-up of the person's hand holding the earphones. The funny aspect of this image is that the person's hand is holding the earphones in the same position as the cell phone is plugged into the charger in the first panel. This creates a humorous visual connection between the two objects, as if the person's hand is plugging the earphones into the charger.

**Our Method:**

The image is a collage of three panels, each showing a different aspect of a cell phone being charged. The first panel features a close-up of the cell phone's charging port, which is plugged into a charger. The second panel shows a person holding a charger, likely preparing to plug it into the cell phone. The third panel displays a person holding a USB cable, which is also being used to charge the cell phone. The funny aspect of this image is that the person in the third panel is holding the USB cable upside down, which is not the correct way to plug it into the cell phone. This adds a humorous element to the otherwise straightforward image of charging a cell phone.

**Figure 23: Case of description task.**

## A.5  GPT-AIDED-EVALUATION TEMPLATE

For gpt-aided-evaluation, we have designed a variety of prompt templates to try to achieve a fairer evaluation. The following is a more effective template for reference.

---

**GPT-aied-evaluation Template**

1. Image Description Evaluation: You will be provided with a set of image descriptions and a list of comments about the image. Your task is to evaluate each comment for hallucinations, which are inaccuracies or inconsistencies with the factual descriptions.

2. Hallucination Identification: Pay special attention to comments that claim the existence of something not present in the descriptions, describe objects or attributes incorrectly, or make unrelated statements.

3. Judgment and Revision: For each comment, provide a judgment (hallucination, correct, or cannot judge) and, if necessary, rewrite the comment to accurately reflect the image content. Ensure that the revised comments are detailed, coherent, and free of hallucinations.

4. Scoring Criteria: Rate the performance of the AI on a scale of 1 to 10 for each of the following criteria:
Accuracy: How well the response aligns with the factual image content.
Detailedness: The richness of the response in necessary details, excluding hallucinated parts.

5. Output Format:
Judgment: List each comment with its judgment (hallucination, correct, or cannot judge) and reason.
Revised Sentences: Provide revised comments where necessary.
Scores: Output the scores for accuracy and detailedness, with reasons.

Example:
Region Descriptions of the Image:
[10, 20, 50, 60]: A red apple on a white plate.
[70, 30, 120, 80]: A blue cup on a wooden table.

Comments for Evaluation:
1. The apple is green.
2. There is a spoon next to the cup.
3. The atmosphere in the room is cozy.

Your Output:

Judgement:
1. hallucination: The description states the apple is red, not green.
2. cannot judge: The region descriptions do not mention a spoon.
3. correct: The comment does not contradict the provided descriptions.

Revised Sentences:
1. The apple is red.
Scores:
Accuracy: 7 8
Reason: Assistant 1 had one hallucination, Assistant 2's response is consistent with the descriptions.
Detailedness: 6 8
Reason: Assistant 1's response lacks necessary details due to the hallucination, Assistant 2 provides a richer description without hallucinations.

---

