# OpenReview forum: "Mitigating Modality Prior-Induced Hallucinations in Multimodal Large Language Models via Deciphering Attention Causality"
_ICLR.cc/2025/Conference — ICLR 2025 Poster_

### Official Review · Reviewer_gJXZ · 2024-10-24

**Soundness:** 2
**Presentation:** 2
**Contribution:** 2
**Rating:** 6
**Confidence:** 3

**Summary:**

This paper introduces a causal inference framework named CausalMM. Considering the influence of visual and textual priors on the predictions of multimodal large language models (MLLMs), the authors employ backdoor adjustment and counterfactual reasoning to mitigate these priors’ effects. Specifically, they design various methods to perturb the attention layers in ViT and LLMs, such as randomization and reversal, to obtain the perturbed model predictions (counterfactual logits). They then use contrastive decoding to enhance the model’s predictive probability distribution. The method is a plug-and-play solution and demonstrates effectiveness on mainstream models and benchmarks.

**Strengths:**

1. This paper presents a simple yet effective method: perturbing the attention matrices in ViT or LLMs to generate counterfactual logits, and then using contrastive decoding to improve the model’s predictive probability distribution.
    - Flexibility and Generalizability. This method is plug-and-play and offers more flexibility compared to VCD, which obtains counterfactual logits by adding noise to images. Perturbing attention allows for greater design space and importantly, can be adapted to modules of different modalities, such as ViT and LLM.
    - Simplicity and Effectiveness. Unlike VCD and its variants that often rely on Adaptive Plausibility Constraints, the proposed method does not appear to require them.
2. The experiments demonstrate the effectiveness of the method.

**Weaknesses:**

1. How was the hyperparameter (e.g., $\gamma$) determined? Were hyperparameters independently tuned for different benchmarks and model/method variants?
2. The choice of base models is not comprehensive. The authors selected LLaVA-1.5 and Qwen2-VL as base models to validate their method, but introducing more models would help confirm the method’s universality. For instance, a table could be added to the appendix to show the method’s cross-model generalizability.
3. Section 3.1 and Figure 2 are hard to follow. Section 3.1 introduces very complex causal relationships, but these seem not closely tied to the method. If I understand correctly, the proposed method is largely similar to VCD: it generates counterfactual logits by perturbation and improves predictions by removing this "background noise" from the normal logits, which does not necessitate such complex preliminary explanations. I would also like to mention that the $P_{effect}$ discussed in Section 3.3 might be unnecessary.
4. The equations are also hard to follow. For example, in Section 3.3, the core of the method involves logits processing, which is unrelated to softmax. Therefore, separating logits from the softmax equation and presenting them independently would be clearer and more concise.

**Questions:**

See Weakness 1.

---

> ### Author Response · Authors · 2024-11-23
> **Author Response to Reviewer gJXZ (Part.1)**
>
> #### We are deeply grateful for the your recognition of our work's innovation and thoroughness, as well as their constructive feedback. We have addressed each suggestion on the manuscript's weaknesses and made the necessary revisions.
>
> > How was the hyperparameter (e.g., $\gamma$) determined? Were hyperparameters independently tuned for different benchmarks and model/method variants?
>
> #### $\gamma$ represents the degree of confidence in the treatment effect and is used to adjust the strength of suppressing the modal prior. We have added this explanation in the text. For different models, the hyperparameter settings that achieve optimal performance are similar.
>
> > The choice of base models is not comprehensive. The authors selected LLaVA-1.5 and Qwen2-VL as base models to validate their method, but introducing more models would help confirm the method’s universality. For instance, a table could be added to the appendix to show the method’s cross-model generalizability.
>
> #### To demonstrate the effectiveness of our approach on large multimodal language models of different architectures, we added experimental data from the Q-former-based InstructBLIP model and the embedding-autoregressive-based Chameleon model to the original experimental data from the vision encoder-mlp-llm paradigm. On the Chameleon model, our method comprehensively surpasses the performance of the baseline model and effectively suppresses the multimodal hallucinations of the model.
>
> Chameleon:
>
> | **Dataset** | **Setting** | **Method**   | **Accuracy** | **Precision** | **Recall** | **F1 Score** |
> | ----------- | ----------- | ------------ | ------------ | ------------- | ---------- | ------------ |
> |             | Random      | Regular      | 61.90        | 57.46         | 91.67      | 70.64        |
> |             |             | **Language** | 69.23        | 63.17         | 92.27      | 74.99        |
> |             |             |              |              |               |            |              |
> | **MSCOCO**  | Popular     | Regular      | 65.10        | 59.86         | 91.67      | 72.43        |
> |             |             | **Language** | 69.43        | 63.34         | 92.27      | 75.12        |
> |             |             |              |              |               |            |              |
> |             | Adversarial | Regular      | 60.20        | 56.28         | 91.40      | 69.66        |
> |             |             | **Language** | 64.00        | 58.94         | 92.33      | 71.95        |
> |             |             |              |              |               |            |              |
> |             | Random      | Regular      | 60.37        | 56.26         | 93.20      | 70.16        |
> |             |             | **Language** | 65.70        | 60.14         | 93.13      | 73.08        |
> |             |             |              |              |               |            |              |
> | **A-OKVQA** | Popular     | Regular      | 57.30        | 54.25         | 93.20      | 68.58        |
> |             |             | **Language** | 63.07        | 58.16         | 93.13      | 71.60        |
> |             |             |              |              |               |            |              |
> |             | Adversarial | Regular      | 53.57        | 51.99         | 93.20      | 66.75        |
> |             |             | **Language** | 56.83        | 53.96         | 93.13      | 68.33        |
> |             |             |              |              |               |            |              |
> |             | Random      | Regular      | 60.37        | 56.26         | 93.20      | 70.16        |
> |             |             | **Language** | 68.43        | 62.18         | 94.13      | 74.89        |
> |             |             |              |              |               |            |              |
> | **GQA**     | Popular     | Regular      | 59.37        | 55.76         | 90.67      | 69.05        |
> |             |             | **Language** | 66.73        | 60.81         | 94.13      | 73.89        |
> |             |             |              |              |               |            |              |
> |             | Adversarial | Regular      | 52.73        | 51.55         | 90.67      | 65.73        |
> |             |             | **Language** | 57.77        | 54.50         | 94.13      | 69.03        |

---

> ### Author Response · Authors · 2024-11-23
> **Author Response to Reviewer gJXZ (Part.2)**
>
> > The choice of base models is not comprehensive. The authors selected LLaVA-1.5 and Qwen2-VL as base models to validate their method, but introducing more models would help confirm the method’s universality. For instance, a table could be added to the appendix to show the method’s cross-model generalizability. (part.2)
>
> #### To demonstrate the effectiveness of our approach on large multimodal language models of different architectures, we added experimental data from the Q-former-based InstructBLIP model and the embedding-autoregressive-based Chameleon model to the original experimental data from the vision encoder-mlp-llm paradigm. On the Chameleon model, our method comprehensively surpasses the performance of the baseline model and effectively suppresses the multimodal hallucinations of the model.
>
> InstructBLIP:
>
> | Dataset   | Setting   | Method     | Accuracy | Precision | Recall | F1 Score |
> |-----------|-----------|------------|----------|-----------|--------|----------|
> |           | Random    | Regular    | 80.71    | 81.67     | 79.19  | 80.41    |
> |           |           | VCD        | 84.53    | 88.55     | 79.32  | 83.68    |
> |           |           | **Vision** | 87.17    | 92.72     | 80.67  | 86.27    |
> |           |           | **Language**| 86.90    | 94.89     | 78.00  | 85.62    |
> |           |           | **Multimodal**| 87.90    | 94.59     | 80.40  | 86.92    |
> | MSCOCO    | Popular   | Regular    | 78.22    | 77.87     | 78.85  | 78.36    |
> |           |           | VCD        | 81.47    | 82.89     | 79.32  | 81.07    |
> |           |           | **Vision** | 83.97    | 86.37     | 80.67  | 83.42    |
> |           |           | **Language**| 83.53    | 87.71     | 78.00  | 82.57    |
> |           |           | **Multimodal**| 84.90    | 88.35     | 80.40  | 84.19    |
> |           | Adversarial| Regular    | 75.84    | 74.30     | 79.03  | 76.59    |
> |           |           | VCD        | 79.56    | 79.67     | 79.39  | 79.52    |
> |           |           | **Vision** | 81.47    | 81.89     | 80.80  | 81.34    |
> |           |           | **Language**| 82.00    | 84.73     | 78.07  | 81.26    |
> |           |           | **Multimodal**| 82.43    | 83.71     | 80.53  | 82.09    |
> |           | Random    | Regular    | 80.91    | 77.97     | 86.16  | 81.86    |
> |           |           | VCD        | 84.11    | 82.21     | 87.05  | 84.56    |
> |           |           | **Vision** | 87.33    | 85.94     | 89.27  | 87.57    |
> |           |           | **Language**| 87.87    | 87.72     | 88.07  | 87.89    |
> |           |           | **Multimodal**| 88.47    | 87.86     | 89.27  | 88.56    |
> | A-OKVQA   | Popular   | Regular    | 76.19    | 72.16     | 85.28  | 78.17    |
> |           |           | VCD        | 79.78    | 76.00     | 87.05  | 81.15    |
> |           |           | **Vision** | 81.07    | 76.69     | 89.27  | 82.50    |
> |           |           | **Language**| 82.33    | 79.01     | 88.07  | 83.29    |
> |           |           | **Multimodal**| 82.13    | 78.45     | 88.60  | 83.22    |
> |           | Adversarial| Regular    | 70.71    | 65.91     | 85.83  | 75.56    |
> |           |           | VCD        | 74.33    | 69.46     | 86.87  | 77.19    |
> |           |           | **Vision** | 74.83    | 69.11     | 89.80  | 78.11    |
> |           |           | **Language**| 76.27    | 71.07     | 88.60  | 78.87    |
> |           |           | **Multimodal**| 75.97    | 70.51     | 89.27  | 78.79    |
> |           | Random    | Regular    | 79.65    | 77.14     | 84.29  | 80.56    |
> |           |           | VCD        | 83.69    | 81.84     | 86.61  | 84.16    |
> |           |           | **Vision** | 86.10    | 84.56     | 88.33  | 86.40    |
> |           |           | **Language**| 86.67    | 86.86     | 86.40  | 86.63    |
> |           |           | **Multimodal**| 87.23    | 86.67     | 88.00  | 87.33    |
> | GQA       | Popular   | Regular    | 73.87    | 69.63     | 84.69  | 76.42    |
> |           |           | VCD        | 78.57    | 74.62     | 86.61  | 80.17    |
> |           |           | **Vision** | 77.77    | 72.92     | 88.33  | 79.89    |
> |           |           | **Language**| 79.17    | 75.48     | 86.40  | 80.57    |
> |           |           | **Multimodal**| 78.97    | 74.99     | 86.93  | 80.52    |
> |           | Adversarial| Regular    | 70.56    | 66.12     | 84.33  | 74.12    |
> |           |           | VCD        | 75.08    | 70.59     | 85.99  | 77.53    |
> |           |           | **Vision** | 74.50    | 69.33     | 87.87  | 77.51    |
> |           |           | **Language**| 76.30    | 71.81     | 86.60  | 78.51    |
> |           |           | **Multimodal**| 75.83    | 71.19     | 86.80  | 78.22    |

---

> ### Author Response · Authors · 2024-11-23
> **Author Response to Reviewer gJXZ (Part.3)**
>
> > Section 3.1 and Figure 2 are hard to follow. Section 3.1 introduces very complex causal relationships, but these seem not closely tied to the method. If I understand correctly, the proposed method is largely similar to VCD: it generates counterfactual logits by perturbation and improves predictions by removing this "background noise" from the normal logits, which does not necessitate such complex preliminary explanations. I would also like to mention that the
> > discussed in Section 3.3 might be unnecessary.
>
> #### We believe that detailed causal modeling can greatly help analyze the connections and independent effects of different factors in the model.
>
> #### Specifically, VCD contrasts outputs derived from original and distorted image inputs. In the contrast, CausalMM isolates the influence of modal priors and other confounders on multimodal attention by using backdoor adjustment methods, obtains the positive treatment effect of attention on output through counterfactual reasoning, adjusts the output of the model at the attention and feature levels, and balances the modality priors. The two are fundamentally different at all levels.
>
> ### Tabular Comparison of CausalMM and VCD
>
> | **Feature**                                                 | **CausalMM**                                                 | **VCD**                                                      |
> | ----------------------------------------------------------- | ------------------------------------------------------------ | ------------------------------------------------------------ |
> | **Core Methodology**                                        | Structural Causal Model (SCM) with backdoor adjustment and counterfactual reasoning | Contrastive decoding                                         |
> | **Focus of Intervention**                                   | Visual and language attention mechanisms, visual features and LLM hidden states | Input image                                                  |
> | **Mechanism of action**                                     | 1.de-confound  2.Obtain the positive treatment effect   3.Adjust attention, features and hidden states   4.Balance the modality priors | Contrasts outputs derived from original and distorted image inputs |
> | **Versatility**                                             | Multimodal hallucinations (vision + language)                | Object hallucinations                                        |
> | **Support single-modal tasks (such as LLM)**                | √                                                            | ×                                                            |
> | **Exploring the causal mechanisms within the model**        | √                                                            | ×                                                            |
> | **Dealing with the confounding effects of modality priors** | √                                                            | ×                                                            |
> | **Modality Priors Addressed**                               | Visual and language priors                                   | -                                                            |
>
>  #### We believe that the discussion of ${P_{effect}}$ in section 3.3 is necessary because they represent the paradigm of counterfactual reasoning in causal theory and fully present the complete process of the do operator and counterfactual reasoning. This is necessary to help readers understand how we obtain the positive causal effect of effective attention on the model output.
>
>
>
> > The equations are also hard to follow. For example, in Section 3.3, the core of the method involves logits processing, which is unrelated to softmax. Therefore, separating logits from the softmax equation and presenting them independently would be clearer and more concise.
>
> #### Thank you for your suggestion. We think the argmax function can more intuitively show our selection process for the next token in the output sequence. If you have more suggestions on the writing format, please let us know.

---

> ### Author Response · Authors · 2024-11-25
> **Summary of response and look forward to the feedback**
>
> #### We greatly appreciate your thoughtful critique and suggestions. Below is a summary of our revisions and clarifications based on your feedback:
>
> - #### **Cross-model generalizability:** As per your request, we provided experimental data in *"Author Response to Reviewer gJXZ (Part 1)"* and *"Author Response to Reviewer gJXZ (Part 2)"* demonstrating the performance of our method on Meta's Chameleon model as well as the InstructBLIP model. We have also included experimental results for **InstructBLIP** and **Chameleon** in the appendix, which confirm that **our method is applicable across several mainstream MLLM architectures**. We encourage you to review these additions.
> - #### **Applicability and generalizability:** Following your suggestions, we discussed hyperparameter-related details in *"Author Response to Reviewer gJXZ (Part 1)"* and added explanations regarding hyperparameters in line 234 of the revised paper. Additionally, we conducted hyperparameter sensitivity tests for our method, with the detailed data provided in *"Author Response to Reviewer CnJZ (Part 3)"*.
> - #### **Differences from the VCD method:** In response to your feedback, we elaborated on the differences between our method and the VCD method in *"Author Response to Reviewer gJXZ (Part 3)"*. Furthermore, we added corresponding content in the appendix of the paper to detail the theoretical derivations related to causal reasoning and reiterated the importance of structural causal modeling.
>
> #### We hope these revisions and clarifications address your concerns and look forward to any additional feedback or questions.

---

> ### Author Response · Authors · 2024-11-27
>
> #### Dear Reviewer gJXZ,
>
> #### Thank you for your valuable time! We are writing to kindly follow up on the status of our manuscript review. We have been actively engaging with other reviewers and have received valuable feedback. Your insights would be greatly appreciated to further enhance the quality of our work.
>
> #### Thank you for your time and consideration!
>
> #### Yours sincerely,
>
> #### CausalMM Team

---

> > ### Comment · Reviewer_gJXZ · 2024-11-27
> > **Response to Authors**
> >
> > Dear Authors,
> >
> > Thank you for your detailed explanation, which thoroughly addressed my concerns and misunderstandings. I also appreciate the updates in the manuscript, which make the paper clearer and more comprehensive. I have updated the rating accordingly.

---

> ### Author Response · Authors · 2024-11-27
>
> #### Dear Reviewer gJXZ,
>
> #### Thank you for your valuable feedback and for taking the time to review our manuscript!
>
> #### Best regards,
>
> #### CausalMM Team

---

### Official Review · Reviewer_vNn2 · 2024-10-30

**Soundness:** 2
**Presentation:** 3
**Contribution:** 3
**Rating:** 6
**Confidence:** 5

**Summary:**

The paper introduces a causal inference framework called CAUSALMM, which is designed to mitigate the issue of multimodal hallucinations in Multimodal Large Language Models (MLLMs) that are often caused by biases from visual and language priors. The core idea is to treat modality priors as a confounder between the attention mechanisms and the model output, and to apply structural causal modeling to address these biases. Specifically, the authors use backdoor adjustment and counterfactual reasoning at both the visual and language attention levels to alleviate the negative effects of modality priors, thereby enhancing the alignment between the MLLM's inputs and outputs.

**Strengths:**

- The paper is well-written and clear motivated.
- The method is plug-and-play and does not require retraining, making it practical for existing MLLMs.
- The integration of causal inference to modify attention weights and optimize token generation in MLLMs is novel.

**Weaknesses:**

- The background knowledge about causal inference is insufficient. The authors do not explain why causal inference is effective in capturing the causal impact of effective attention in MLLM output.
- Several claims lack explanations and references. The authors are advised to carefully proofread the paper and add the necessary citations. For instance:
    - Line 65: “Modality priors are one of the confounding factors in the causal path of MLLM.” Why? And what is the definition of the confounding factor?
    - Line 58-61: “existing decoding strategies……, overlooking the causal relationships among attention visual attention, language attention, modality priors, and model output.”
- In Section 3.3, the authors introduce the causal effect and the corresponding calculation of selecting the next token. But what’s next after determining the index of the next token? The authors are advised to supplement more on this point. I assume that it is going to talk about inferencing with the modified next token prediction.
- Though the proposed method is somewhat novel, the experimental results are not quite significant and robust compared with existing methods (Table 1).
- Lack of implementation details. It would be more convincing to provide more details on how to reproduce the experimental results.
- The notations should be considered more carefully. For example, in Section 3.3, it is confusing that in Line 222-223 $O$ represents the original output but in Line 226-227 it denotes the model output.

**Questions:**

- In Lines 58-61, the authors state that existing decoding strategies overlook the causal relationships among attention visual attention, language attention, modality priors, and model output. However, VCD [1] injects noise into the image input and acquires the output logits for contrastive decoding, which can be seen as a counterfactual intervention at the visual input. The effects of this intervention are also later shown in the attention parts and model outputs [2,3]. Could the authors make more discussion on this perspective?
- What does $\gamma$ mean in the equations in Section 3.3? More importantly, the calculation of choosing the index of the next token is not discussed. Why is it calculated in this way?
- Could you provide the results of VCD and OPERA on VLind-Bench?

[1] Leng S, Zhang H, Chen G, et al. Mitigating object hallucinations in large vision-language models through visual contrastive decoding[C]//Proceedings of the IEEE/CVF Conference on Computer Vision and Pattern Recognition. 2024: 13872-13882.

[2] Xiao X, Wu B, Wang J, et al. Seeing the Image: Prioritizing Visual Correlation by Contrastive Alignment[J]. arXiv preprint arXiv:2405.17871, 2024.

[3] Chen Z, Xu C, Qi Y, et al. Mllm is a strong reranker: Advancing multimodal retrieval-augmented generation via knowledge-enhanced reranking and noise-injected training[J]. arXiv preprint arXiv:2407.21439, 2024.

---

> ### Author Response · Authors · 2024-11-23
> **Author Response to Reviewer vNn2 (Part.1)**
>
> #### We are deeply grateful for your recognition of our work's innovation and thoroughness, as well as their constructive feedback. We have addressed each suggestion on the manuscript's weaknesses and made the necessary revisions.
>
> > Insufficient Theoretical Justification: The paper lacks a deep theoretical analysis of why the proposed causal interventions lead to improved performance. The causal model is described, but the theoretical foundations and assumptions are not thoroughly explored or justified.
>
> #### We have supplemented some arguments related to the causal inference theory in the paper. We will describe how our causal inference method balances the model's modal priors from three aspects: structural causal model, backdoor adjustment, and counterfactual reasoning:
>
> ## Structural Causal Model (SCM)
>
> #### The process of methods based on causal theory includes building the structural causal models and choosing the proper way of causal inference. In the structure, nodes indicate the important variables or variables that are hidden but have an effect on the important ones, while the edges indicate inner causal relationships. Combining different causal structures and different concerns, suitable causal reasoning methods are selected to eliminate spurious correlations.
>
> ## Backdoor Adjustment
>
> #### In this work, we reinterpret the backdoor adjustment framework to analyze the causal influence of modality priors on attention mechanisms and model outputs. By identifying modality priors ($M$) as confounders, we isolate the causal effect of attention ($A$) on the output ($O$) using the backdoor adjustment method.
>
> ####  **Variables:**
> ####  $A$ (attention): The mechanism whose causal effect we aim to evaluate.
> ####  $M$ (modality priors): Influences both $A$ and $O$, acting as a confounder.
> ####  $O$ (model output): The outcome variable influenced by $A$ and $M$.
>
> ####  **Causal Challenge:**
> ####   The backdoor path $A \leftarrow M \to O$ introduces confounding, making it necessary to adjust for $M$ to isolate the causal effect of $A$ on $O$.
>
> #### To block confounding, the backdoor criterion ensures that:
>
> #### 1. $M$ blocks all backdoor paths from $A$ to $O$.
> #### 2. $M$ is not influenced by $A$.
>
> #### Using this criterion, the causal effect of $A$ on $O$ is computed as
>
> $
> P(o \mid do(a)) = \sum_m P(o \mid a, m) P(m),
> $
>
> #### Modality priors ($M$) explain the indirect influence of $A$ on $O$, enabling a disentangled analysis. Adjusting for $M$ removes the confounding, ensuring that $A$'s causal impact on $O$ is properly estimated.
>
> ## Counterfactual reasoning
>
> #### By controlling for confounders, we can more accurately estimate causal relationships. This method serves as a foundation for counterfactual reasoning, which enables the assessment of treatment effects in systems like multimodal models.
>
> ### Causal Effect of Visual Attention ($A_i$)
>
> #### The causal effect of the visual attention mechanism on the model output $O$ is given by:
>
> $
> P_{effect\_V} = E_{A_i \sim \tilde{A}_i}\left[P(O | A_i = \textbf{A}_i, I = \textbf{I}, P_v = \textbf{P}_v) - P(O | \text{do}(A_i = \textbf{a}_i), I = \textbf{I}, P_v = \textbf{P}_v)\right].
> $
>
> #### Here:
>
> #### $P_{effect\_V}$ represents the treatment effect of visual attention on the output $O$.
> ####  $\textbf{A}_i$ denotes the observed visual attention, while $\textbf{a}_i$ represents the intervention applied to the visual attention.
>
> ### Causal Effect of Language Model Attention ($A_t$)
>
> #### Similarly, the causal effect of the language model attention on the output $O$ can be expressed as:
>
> $
> P_{effect\_L} = E_{A_t \sim \tilde{A}_t}\left[P(O | A_t = \textbf{A}_t, T_t = \textbf{T}_t, P_l = \textbf{P}_l) - P(O | \text{do}(A_t = \textbf{a}_t), T_t = \textbf{T}_t, P_l = \textbf{P}_l)\right],
> $
>
> #### Here:
>
> ####  $P_{effect\_L}$ represents the treatment effect of language attention on the output $O$.
> ####  $\textbf{A}_t$ is the observed language model attention, while $\textbf{a}_t$ represents the intervention on the language model attention.
>
> ### Combined Causal Effect in a Multimodal Setting
>
> #### In multimodal systems, the combined treatment effect of both visual and language attention mechanisms is described as:
>
> $
> P_{effect\_M} = E_{A_i, A_t \sim \tilde{A}_i, \tilde{A}_t}\left[P(O | A_i = \textbf{A}_i, A_t = \textbf{A}_t, I = \textbf{I}, T_t = \textbf{T}_t, P_v = \textbf{P}_v, P_l = \textbf{P}_l)\right]  - P(O | \text{do}(A_i = \textbf{a}_i), \text{do}(A_t = \textbf{a}_t), I = \textbf{I}, T_t = \textbf{T}_t, P_v = \textbf{P}_v, P_l = \textbf{P}_l),
> $
>
> #### In this formulation:
>
> ####  $P_{effect\_M}$ measures the combined effect of both visual and language attention mechanisms on the model output.
> ####  The observed and intervened attention variables are denoted by $\textbf{A}_i, \textbf{a}_i$ for visual attention, and $\textbf{A}_t, \textbf{a}_t$ for language attention.

---

> ### Author Response · Authors · 2024-11-23
> **Author Response to Reviewer vNn2 (Part.2)**
>
> > Several claims lack explanations and references. The authors are advised to carefully proofread the paper and add the necessary citations. For instance:
> > Line 65: “Modality priors are one of the confounding factors in the causal path of MLLM.” Why? And what is the definition of the confounding factor?
> > Line 58-61: “existing decoding strategies……, overlooking the causal relationships among attention visual attention, language attention, modality priors, and model output.
>
> #### We added references to relevant articles to support our point of view. At the same time, for the possible confusion, we modified the sentence in the original text to make the conclusion more moderate.
>
> #### **Confounding Factor**: A confounding factor, also known as a confounder, is a variable that influences both the dependent variable and independent variable, causing a spurious association. In simpler terms, it's an outside influence that can distort the true relationship between the variables being studied.
>
> #### Without causal inference theory, the true impact of factors such as attention on model output under the influence of modal priors has not been well studied.
>
> #### [1] Peng, D., Wei, W., Mao, X., Fu, Y., Chen, D. "An Empirical Study on the Language Modal in Visual Question Answering". arXiv preprint arXiv:2305.10143
>
> #### [2] Chen, M., Cao, Y., Zhang, Y., Lu, C. "Quantifying and Mitigating Unimodal Biases in Multimodal Large Language Models: A Causal Perspective". arXiv preprint arXiv:2403.18346
>
> #### [3] Lukics, K. S., Lukács, Á. "Modality, presentation, domain and training effects in statistical learning". Sci Rep, 2022.
>
> #### [4] Gema, A. P., Jin, C., Abdulaal, A., Diethe, T., Teare, P., Alex, B., Minervini, P., Saseendran, A. "DeCoRe: Decoding by Contrasting Retrieval Heads to Mitigate Hallucinations". arXiv preprint arXiv:2410.18860
>
> #### [5] Lee K, Kim M, Yoon S, et al. VLind-Bench: Measuring Language Priors in Large Vision-Language Models. arXiv preprint arXiv:2406.08702, 2024.
>
> > In Section 3.3, the authors introduce the causal effect and the corresponding calculation of selecting the next token. But what’s next after determining the index of the next token? The authors are advised to supplement more on this point. I assume that it is going to talk about inferencing with the modified next token prediction.
>
> #### We have added content about the selection of the next token in the revised version. We use direct sampling as a decoding strategy because it can improve the diversity of text and reduce the probability of repeatedly outputting the same word.
>
> > Lack of implementation details. It would be more convincing to provide more details on how to reproduce the experimental results.
>
> #### We are already preparing an open source version of the code to increase readability. Please pay attention to our follow-up work.
>
>
> > The notations should be considered more carefully. For example, in Section 3.3, it is confusing that in Line 222-223 represents the original output but in Line 226-227 it denotes the model output.
>
> #### Thank you for your feedback. We have corrected the representation of the relevant variables in the revised version.
>
> > What does $\gamma$ mean in the equations in Section 3.3?
>
> #### $\gamma$ represents the degree of confidence in the treatment effect and is used to adjust the strength of suppressing the modal prior. We have added this explanation in the text.
>
>
> > Could you provide the results of VCD and OPERA on VLind-Bench?
>
> #### We provide experimental data of VCD and OPERA methods on the VLind benchmark. The result of the OPERA method is 0, which is similar to some models in Table 1 in the original VLind paper[1]. No conclusion is given in the original paper. The reason is speculated to be the decline in the ability to follow certain instructions.
>
> | **Metrics**  | **$S_{ck}$** | **$S_{vp}$** | **$S_{cb}$** | **$S_{lp}$** | **CB** | **LP** |
> | ------------ | ------------ | ------------ | ------------ | ------------ | ------ | ------ |
> | **Regular**  | 32.1         | 40.7         | 43.3         | 33.1         | 43.7   | 27.1   |
> | **VCD**      | 30.5         | 48.0         | 47.8         | 31.0         | 44.0   | 29.2   |
> | **OPERA***   | 0            | 0            | -            | -            | 0      | 0      |
> | **CausalMM** | 57.0         | 80.8         | 64.0         | 61.8         | 59.9   | 40.2   |
>
> #### [1] Lee K, Kim M, Yoon S, et al. VLind-Bench: Measuring Language Priors in Large Vision-Language Models[J]. arXiv preprint arXiv:2406.08702, 2024.

---

> > ### Author Response · Authors · 2024-11-23
> > **Author Response to Reviewer vNn2 (Part.4)**
> >
> > > Though the proposed method is somewhat novel, the experimental results are not quite significant and robust compared with existing methods (Table 1).
> >
> > #### We added this table to expand the comparison with more baselines. In the table, the values taken are the averages of the three parts of the POPE benchmark (MSCOCO, A-OKVQA, GQA). It can be seen that the CausalMM method can achieve the highest value most of the time.
> >
> > | Dataset         | Setting     | Method   | Accuracy  | Precision  | Recall  | F1 Score  |
> > |-----------------|-------------|----------|-----------|------------|---------|-----------|
> > |                 | Random      | Regular  | 80.42     | 78.93      | 83.21   | 80.94     |
> > |                 |    | DOLA     | 83.00     | 83.06      | 83.13   | 83.00     |
> > |                 |     | VCD      | 84.11     | 84.20      | 84.33   | 84.13     |
> > |                 |     | OPERA    | 85.07     | 88.39      | 80.73   | 84.39     |
> > |                 |        | AGLA     | 87.30     | 88.83      | 85.68   | 87.07     |
> > |                 |  | Vision   | 86.87     | 87.74      | 86.09   | 86.75     |
> > |                 |    | Language | 87.15     | 89.82      | 84.16   | 86.71     |
> > |                 |   | Multimodal| 87.87     | 89.71      | 85.89   | 87.60     |
> > | InstructBLIP    | Popular     | Regular  | 76.09     | 73.22      | 82.94   | 77.65     |
> > |                 |       | DOLA     | 78.99     | 77.12      | 83.13   | 79.85     |
> > |                 |       | VCD      | 79.94     | 77.84      | 84.33   | 80.80     |
> > |    |       | OPERA    | 78.33     | 73.85      | 87.73   | 80.20     |
> > |      |      | AGLA     | 81.86     | 80.17      | 85.68   | 82.58     |
> > |     |      | Vision   | 80.94     | 78.66      | 86.09   | 81.94     |
> > |      |             | Language | 81.68     | 80.73      | 84.16   | 82.14     |
> > |         |             | Multimodal| 82.00     | 80.60      | 85.31   | 82.64     |
> > |      | Adversarial | Regular  | 72.37     | 68.78      | 83.06   | 75.42     |
> > |   |             | DOLA     | 74.67     | 71.53      | 83.11   | 76.68     |
> > |   |             | VCD      | 76.32     | 73.24      | 84.08   | 78.08     |
> > |   |             | OPERA    | 75.50     | 70.49      | 87.73   | 78.17     |
> > |   |             | AGLA     | 77.29     | 74.09      | 85.67   | 79.16     |
> > |           |       | Vision   | 76.93     | 73.44      | 86.16   | 78.99     |
> > |                 |             | Language | 78.19     | 75.87      | 84.42   | 79.55     |
> > |    |             | Multimodal| 78.08     | 75.14      | 85.53   | 79.70     |
> > |   | Random      | Regular  | 83.72     | 89.30      | 77.13   | 82.55     |
> > |   |             | DOLA     | 84.78     | 87.59      | 81.27   | 84.19     |
> > |                 |             | VCD      | 86.05     | 90.39      | 80.91   | 85.29     |
> > |                 |             | OPERA    | 88.64     | 88.09      | 89.73   | 87.43     |
> > |                 |             | AGLA     | 88.54     | 94.41      | 82.08   | 87.71     |
> > |   |             | Vision   | 87.17     | 92.35      | 81.28   | 86.33     |
> > |  |             | Language | 86.84     | 91.96      | 80.86   | 85.68     |
> > |    |             | Multimodal| 88.79     | 92.63      | 84.35   | 88.26     |
> > | LLaVA-1.5       | Popular     | Regular  | 79.73     | 82.03      | 76.73   | 79.11     |
> > |    |             | DOLA     | 79.75     | 84.11      | 76.22   | 80.61     |
> > |                 |             | VCD      | 81.52     | 82.59      | 80.60   | 81.39     |
> > |   |             | OPERA    | 83.34     | 80.27      | 89.73   | 84.44     |
> > |    |             | AGLA     | 85.14     | 87.88      | 82.08   | 84.68     |
> > |   |             | Vision   | 83.13     | 84.84      | 81.37   | 82.85     |
> > |     |             | Language | 84.31     | 86.75      | 83.80   | 84.26     |
> > |      |             | Multimodal| 85.06     | 86.44      | 83.82   | 84.87     |
> > |                 | Adversarial | Regular  | 76.02     | 76.20      | 76.60   | 76.36     |
> > |                 |             | DOLA     | 76.32     | 77.27      | 75.47   | 76.16     |
> > |                 |             | VCD      | 77.84     | 76.87      | 80.75   | 78.53     |
> > |                 |             | OPERA    | 76.68     | 71.66      | 89.71   | 79.46     |
> > |                 |             | AGLA     | 81.13     | 81.20      | 82.10   | 81.36     |
> > |                 |             | Vision   | 78.62     | 77.83      | 81.51   | 79.31     |
> > |                 |             | Language | 78.59     | 78.49      | 79.77   | 78.90     |
> > |                 |             | Multimodal| 80.36     | 79.53      | 82.86   | 80.91     |

---

> > > ### Comment · Reviewer_vNn2 · 2024-11-25
> > >
> > > Thanks a lot for the authors' response and care for my concerns. I have updated the rating accordingly.

---

> > > > ### Author Response · Authors · 2024-11-25
> > > >
> > > > Thank you very much for your kind response. We truly appreciate your thoughtful feedback and support!

---

> ### Author Response · Authors · 2024-11-23
> **Author Response to Reviewer vNn2 (Part.3)**
>
> > In Lines 58-61, the authors state that existing decoding strategies overlook the causal relationships among attention visual attention, language attention, modality priors, and model output. However, VCD [1] injects noise into the image input and acquires the output logits for contrastive decoding, which can be seen as a counterfactual intervention at the visual input. The effects of this intervention are also later shown in the attention parts and model outputs [2,3]. Could the authors make more discussion on this perspective?
>
> #### Thank you for your feedback. Specifically, VCD contrasts outputs derived from original and distorted image inputs. In the contrast, CausalMM isolates the influence of modal priors and other confounders on multimodal attention by using backdoor adjustment methods, obtains the positive treatment effect of attention on output through counterfactual reasoning, adjusts the output of the model at the attention and feature levels, and balances the modality priors. The former's operation is not consistent with the definition of intervention in the causal theory system. Causal reasoning does not directly use intervention to optimize the output.
>
> ### Tabular Comparison of CausalMM and VCD
>
> | **Feature**                                                 | **CausalMM**                                                 | **VCD**                                                      |
> | ----------------------------------------------------------- | ------------------------------------------------------------ | ------------------------------------------------------------ |
> | **Core Methodology**                                        | Structural Causal Model (SCM) with backdoor adjustment and counterfactual reasoning | Contrastive decoding                                         |
> | **Focus of Intervention**                                   | Visual and language attention mechanisms, visual features and LLM hidden states | Input image                                                  |
> | **Mechanism of action**                                     | 1.de-confound  2.Obtain the positive treatment effect   3.Adjust attention, features and hidden states   4.Balance the modality priors | Contrasts outputs derived from original and distorted image inputs |
> | **Versatility**                                             | Multimodal hallucinations (vision + language)                | Object hallucinations                                        |
> | **Support single-modal tasks (such as LLM)**                | √                                                            | ×                                                            |
> | **Exploring the causal mechanisms within the model**        | √                                                            | ×                                                            |
> | **Dealing with the confounding effects of modality priors** | √                                                            | ×                                                            |
> | **Modality Priors Addressed**                               | Visual and language priors                                   | -                                                            |

---

### Official Review · Reviewer_CnJZ · 2024-10-31

**Soundness:** 3
**Presentation:** 3
**Contribution:** 2
**Rating:** 5
**Confidence:** 4

**Summary:**

This paper addresses the problem of modality prior-induced hallucinations in Multimodal Large Language Models (MLLMs). The authors propose a causal reasoning framework, which applies structural causal modeling and counterfactual reasoning to MLLMs. By treating modality priors as confounding factors between attention mechanisms and the model's output, the approach aims to mitigate the negative effects of these priors. The method involves interventions on both visual and language attention mechanisms and is evaluated on several benchmarks.

**Strengths:**

1. **Perspective**: The paper introduces the idea of applying causal inference techniques to address modality prior-induced hallucinations in MLLMs, which is an interesting and potentially valuable perspective.

2. **Comprehensive Experiments**: The authors conduct experiments on multiple benchmarks, including VLind-Bench, POPE, and MME, providing a range of evaluations for their method.

3. **Ablation Studies**: Inclusion of ablation studies helps in understanding the impact of different components of the proposed method.

**Weaknesses:**

1. **Insufficient Theoretical Justification**: The paper lacks a deep theoretical analysis of why the proposed causal interventions lead to improved performance. The causal model is described, but the theoretical foundations and assumptions are not thoroughly explored or justified.

2. **Limited Novelty**: While the application of causal inference to MLLMs is presented as novel, causal reasoning has been previously applied in machine learning models, including language models. The paper does not sufficiently differentiate its contributions from existing work in causal inference applied to deep learning.

3. **Inadequate Comparison with Baselines**: For some evaluations(such as figure 3,5,6), the experimental evaluation compares the proposed method with only a partial set of baselines from the setup. Under many settings, the performance of this method underperform or only slightly outperform baseline methods as shown in table 1.

4. **Superficial Experimental Analysis**: The results, while showing improvements, lack statistical significance testing. Additionally, there is a lack of detailed analysis of where and why the method improves performance, making it difficult to assess the true impact.

5. **Applicability and Generalization**: The approach is tested on specific MLLMs, but it is unclear how well the method generalizes to other models.

6. **Lack of Discussion on Limitations**: The paper does not adequately discuss the limitations or potential downsides of the proposed method, such as scenarios where it might not work well or possible negative impacts.

**Questions:**

1. **Theoretical Justification**: Can you provide a more detailed theoretical analysis or proofs to support the efficacy of your causal interventions?

2. **Computational Complexity**: What is the computational overhead introduced by your method? How does it compare to the baseline models in terms of runtime and resource consumption?

3. **Generality of the Approach**: How well does your method generalize to other types of MLLMs, such as chameleon?

4. **Limitations and Failure Cases**: What are the limitations of your method? Are there scenarios where it does not perform well or might even degrade performance? How do you address potential negative impacts?

5. **Impact of Hyperparameters**: How sensitive is your method to the choice of hyperparameters involved in the interventions? Have you performed a sensitivity analysis?

6. **Robustness to Noise**: How does your method handle noisy or adversarial inputs? Does the causal framework improve robustness in such cases?

---

> ### Author Response · Authors · 2024-11-23
> **Author Response to Reviewer CnJZ (Part.1)**
>
> #### We are deeply grateful for your recognition of our work's innovation and thoroughness, as well as their constructive feedback. We have addressed each suggestion on the manuscript's weaknesses and made the necessary revisions.
>
> > Insufficient Theoretical Justification: The paper lacks a deep theoretical analysis of why the proposed causal interventions lead to improved performance. The causal model is described, but the theoretical foundations and assumptions are not thoroughly explored or justified.
>
> #### We have supplemented some arguments related to the causal inference theory in the paper. We will describe how our causal inference method balances the model's modal priors from three aspects: structural causal model, backdoor adjustment, and counterfactual reasoning:
>
> ## Structural Causal Model (SCM)
>
> #### The process of methods based on causal theory includes building the structural causal models and choosing the proper way of causal inference. In the structure, nodes indicate the important variables or variables that are hidden but have an effect on the important ones, while the edges indicate inner causal relationships. Combining different causal structures and different concerns, suitable causal reasoning methods are selected to eliminate spurious correlations.
>
> ## Backdoor Adjustment
>
> #### In this work, we reinterpret the backdoor adjustment framework to analyze the causal influence of modality priors on attention mechanisms and model outputs. By identifying modality priors ($M$) as confounders, we isolate the causal effect of attention ($A$) on the output ($O$) using the backdoor adjustment method.
>
> ####  **Variables:**
> ####  $A$ (attention): The mechanism whose causal effect we aim to evaluate.
> ####  $M$ (modality priors): Influences both $A$ and $O$, acting as a confounder.
> ####  $O$ (model output): The outcome variable influenced by $A$ and $M$.
>
> ####  **Causal Challenge:**
> ####   The backdoor path $A \leftarrow M \to O$ introduces confounding, making it necessary to adjust for $M$ to isolate the causal effect of $A$ on $O$.
>
> #### To block confounding, the backdoor criterion ensures that:
>
> #### 1. $M$ blocks all backdoor paths from $A$ to $O$.
> #### 2. $M$ is not influenced by $A$.
>
> #### Using this criterion, the causal effect of $A$ on $O$ is computed as
>
> $
> P(o \mid do(a)) = \sum_m P(o \mid a, m) P(m),
> $
>
> #### Modality priors ($M$) explain the indirect influence of $A$ on $O$, enabling a disentangled analysis. Adjusting for $M$ removes the confounding, ensuring that $A$'s causal impact on $O$ is properly estimated.
>
> ## Counterfactual reasoning
>
> #### By controlling for confounders, we can more accurately estimate causal relationships. This method serves as a foundation for counterfactual reasoning, which enables the assessment of treatment effects in systems like multimodal models.
>
> ### Causal Effect of Visual Attention ($A_i$)
>
> #### The causal effect of the visual attention mechanism on the model output $O$ is given by:
>
> $
> P_{effect\_V} = E_{A_i \sim \tilde{A}_i}\left[P(O | A_i = \textbf{A}_i, I = \textbf{I}, P_v = \textbf{P}_v) - P(O | \text{do}(A_i = \textbf{a}_i), I = \textbf{I}, P_v = \textbf{P}_v)\right].
> $
>
> #### Here:
>
> #### $P_{effect\_V}$ represents the treatment effect of visual attention on the output $O$.
> ####  $\textbf{A}_i$ denotes the observed visual attention, while $\textbf{a}_i$ represents the intervention applied to the visual attention.
>
> ### Causal Effect of Language Model Attention ($A_t$)
>
> #### Similarly, the causal effect of the language model attention on the output $O$ can be expressed as:
>
> $
> P_{effect\_L} = E_{A_t \sim \tilde{A}_t}\left[P(O | A_t = \textbf{A}_t, T_t = \textbf{T}_t, P_l = \textbf{P}_l) - P(O | \text{do}(A_t = \textbf{a}_t), T_t = \textbf{T}_t, P_l = \textbf{P}_l)\right],
> $
>
> #### Here:
>
> ####  $P_{effect\_L}$ represents the treatment effect of language attention on the output $O$.
> ####  $\textbf{A}_t$ is the observed language model attention, while $\textbf{a}_t$ represents the intervention on the language model attention.
>
> ### Combined Causal Effect in a Multimodal Setting
>
> #### In multimodal systems, the combined treatment effect of both visual and language attention mechanisms is described as:
>
> $
> P_{effect\_M} = E_{A_i, A_t \sim \tilde{A}_i, \tilde{A}_t}\left[P(O | A_i = \textbf{A}_i, A_t = \textbf{A}_t, I = \textbf{I}, T_t = \textbf{T}_t, P_v = \textbf{P}_v, P_l = \textbf{P}_l)\right]  - P(O | \text{do}(A_i = \textbf{a}_i), \text{do}(A_t = \textbf{a}_t), I = \textbf{I}, T_t = \textbf{T}_t, P_v = \textbf{P}_v, P_l = \textbf{P}_l),
> $
>
> #### In this formulation:
>
> ####  $P_{effect\_M}$ measures the combined effect of both visual and language attention mechanisms on the model output.
> ####  The observed and intervened attention variables are denoted by $\textbf{A}_i, \textbf{a}_i$ for visual attention, and $\textbf{A}_t, \textbf{a}_t$ for language attention.

---

> ### Author Response · Authors · 2024-11-23
> **Author Response to Reviewer CnJZ (Part.2)**
>
> > Limited Novelty: While the application of causal inference to MLLMs is presented as novel, causal reasoning has been previously applied in machine learning models, including language models. The paper does not sufficiently differentiate its contributions from existing work in causal inference applied to deep learning.
>
> #### While causal inference has been applied in machine learning, the novelty of CausalMM lies in its specifically designed structural causal model and counterfactual reasoning framework tailored for Multimodal Large Language Models (MLLMs). Unlike previous works, CausalMM explicitly addresses the causal relationships between visual and language modalities and identifies and adjusts the influence of modality priors on attention mechanisms through intervention.
>
> | **Feature**           | **Focus**                                                    | **Empirical**                                                | **Innovation**                                               | **Technique**                                                |
> | --------------------- | ------------------------------------------------------------ | ------------------------------------------------------------ | ------------------------------------------------------------ | ------------------------------------------------------------ |
> | **Rhino** [1]         | Causal relationship learning from time-series data           | Extensive synthetic experiments and real-world benchmarks    | Combining vector auto-regression, deep learning, and variational inference | Modeling non-linear relationships with history-dependent noise and instantaneous effects |
> | **Causal-StoNet** [2] | High-dimensional complex data                                | Extensive numerical studies                                  | Sparse deep learning theory                                  | Adaptive Stochastic Gradient MCMC (SGMCMC)                   |
> | **CUTS** [3]          | Causal discovery from irregular time-series data             | Joint imputation of unobserved data points and causal graph construction | Delayed Supervision Graph Neural Network (DSGNN) for unstructured data | Iterative framework with mutually boosting modules for data prediction and graph fitting |
> | **VBCI** [4]          | Predicting cellular gene expressions under counterfactual perturbations | Extensive experiments demonstrating superiority over state-of-the-art deep learning models | Novel graph variational Bayesian causal inference framework utilizing gene regulatory networks | Adjacency matrix updating for graph convolutional networks during pre-training |
> | CausalMM (Ours)       | Mitigating multimodal hallucinations in MLLMs                | Extensive experiments on VLind-Bench, POPE, and MME benchmarks | Causal inference framework (CausalMM) for attention mechanism | Back-door adjustment and counterfactual reasoning at attention levels |
>
> #### [1] Gong, W., Jennings, J., Zhang, C., Pawlowski, N. "Rhino: Deep Causal Temporal Relationship Learning With History-dependent Noise". Machine Learning, 2022.
>
> #### [2] Fang, Y., Liang, F. "Causal-StoNet: Causal Inference for High-Dimensional Complex Data". arXiv preprint arXiv:2403.18994
>
> #### [3] Cheng, Y., He, K., Xiao, T., Dai, Q., Suo, J., Li, Z., Yang, R. "CUTS: Neural Causal Discovery from Irregular Time-Series Data". International Conference on Learning Representations, 2023.
>
> #### [4] Voloch, L., Barton, R. A., Ioannidis, V., Donno, C. D., Wu, Y., Price, L., Karypis, G., Wang, Z. "Predicting Cellular Responses with Variational Causal Inference and Refined Relational Information". International Conference on Learning Representations, 2022.

---

> ### Author Response · Authors · 2024-11-23
> **Author Response to Reviewer CnJZ (Part.3)**
>
> > Inadequate Comparison with Baselines: For some evaluations(such as figure 3,5,6), the experimental evaluation compares the proposed method with only a partial set of baselines from the setup. Under many settings, the performance of this method underperform or only slightly outperform baseline methods as shown in table 1.
>
> #### We added this table to expand the comparison with more baselines. In the table, the values taken are the averages of the three parts of the POPE benchmark (MSCOCO, A-OKVQA, GQA). It can be seen that the CausalMM method can achieve the highest value most of the time.
>
> | Dataset         | Setting     | Method   | Accuracy  | Precision  | Recall  | F1 Score  |
> |-----------------|-------------|----------|-----------|------------|---------|-----------|
> |                 | Random      | Regular  | 80.42     | 78.93      | 83.21   | 80.94     |
> |                 |    | DOLA     | 83.00     | 83.06      | 83.13   | 83.00     |
> |                 |     | VCD      | 84.11     | 84.20      | 84.33   | 84.13     |
> |                 |     | OPERA    | 85.07     | 88.39      | 80.73   | 84.39     |
> |                 |        | AGLA     | 87.30     | 88.83      | 85.68   | 87.07     |
> |                 |  | Vision   | 86.87     | 87.74      | 86.09   | 86.75     |
> |                 |    | Language | 87.15     | 89.82      | 84.16   | 86.71     |
> |                 |   | Multimodal| 87.87     | 89.71      | 85.89   | 87.60     |
> | InstructBLIP    | Popular     | Regular  | 76.09     | 73.22      | 82.94   | 77.65     |
> |                 |       | DOLA     | 78.99     | 77.12      | 83.13   | 79.85     |
> |                 |       | VCD      | 79.94     | 77.84      | 84.33   | 80.80     |
> |    |       | OPERA    | 78.33     | 73.85      | 87.73   | 80.20     |
> |      |      | AGLA     | 81.86     | 80.17      | 85.68   | 82.58     |
> |     |      | Vision   | 80.94     | 78.66      | 86.09   | 81.94     |
> |      |             | Language | 81.68     | 80.73      | 84.16   | 82.14     |
> |         |             | Multimodal| 82.00     | 80.60      | 85.31   | 82.64     |
> |      | Adversarial | Regular  | 72.37     | 68.78      | 83.06   | 75.42     |
> |   |             | DOLA     | 74.67     | 71.53      | 83.11   | 76.68     |
> |   |             | VCD      | 76.32     | 73.24      | 84.08   | 78.08     |
> |   |             | OPERA    | 75.50     | 70.49      | 87.73   | 78.17     |
> |   |             | AGLA     | 77.29     | 74.09      | 85.67   | 79.16     |
> |           |       | Vision   | 76.93     | 73.44      | 86.16   | 78.99     |
> |                 |             | Language | 78.19     | 75.87      | 84.42   | 79.55     |
> |    |             | Multimodal| 78.08     | 75.14      | 85.53   | 79.70     |
> |   | Random      | Regular  | 83.72     | 89.30      | 77.13   | 82.55     |
> |   |             | DOLA     | 84.78     | 87.59      | 81.27   | 84.19     |
> |                 |             | VCD      | 86.05     | 90.39      | 80.91   | 85.29     |
> |                 |             | OPERA    | 88.64     | 88.09      | 89.73   | 87.43     |
> |                 |             | AGLA     | 88.54     | 94.41      | 82.08   | 87.71     |
> |   |             | Vision   | 87.17     | 92.35      | 81.28   | 86.33     |
> |  |             | Language | 86.84     | 91.96      | 80.86   | 85.68     |
> |    |             | Multimodal| 88.79     | 92.63      | 84.35   | 88.26     |
> | LLaVA-1.5       | Popular     | Regular  | 79.73     | 82.03      | 76.73   | 79.11     |
> |    |             | DOLA     | 79.75     | 84.11      | 76.22   | 80.61     |
> |                 |             | VCD      | 81.52     | 82.59      | 80.60   | 81.39     |
> |   |             | OPERA    | 83.34     | 80.27      | 89.73   | 84.44     |
> |    |             | AGLA     | 85.14     | 87.88      | 82.08   | 84.68     |
> |   |             | Vision   | 83.13     | 84.84      | 81.37   | 82.85     |
> |     |             | Language | 84.31     | 86.75      | 83.80   | 84.26     |
> |      |             | Multimodal| 85.06     | 86.44      | 83.82   | 84.87     |
> |                 | Adversarial | Regular  | 76.02     | 76.20      | 76.60   | 76.36     |
> |                 |             | DOLA     | 76.32     | 77.27      | 75.47   | 76.16     |
> |                 |             | VCD      | 77.84     | 76.87      | 80.75   | 78.53     |
> |                 |             | OPERA    | 76.68     | 71.66      | 89.71   | 79.46     |
> |                 |             | AGLA     | 81.13     | 81.20      | 82.10   | 81.36     |
> |                 |             | Vision   | 78.62     | 77.83      | 81.51   | 79.31     |
> |                 |             | Language | 78.59     | 78.49      | 79.77   | 78.90     |
> |                 |             | Multimodal| 80.36     | 79.53      | 82.86   | 80.91     |

---

> ### Author Response · Authors · 2024-11-23
> **Author Response to Reviewer CnJZ (Part.4)**
>
> > Superficial Experimental Analysis: The results, while showing improvements, lack statistical significance testing. Additionally, there is a lack of detailed analysis of where and why the method improves performance, making it difficult to assess the true impact.
>
> #### We performed a statistical significance analysis on the experimental results. The significance level is 0.05. The results of the statistical significance test support the advantages of our method and prove its effectiveness in practical applications.
>
> | **Metric** | **t-statistic** | **p-value** | **Significant** |
> | ---------- | --------------- | ----------- | --------------- |
> | Accuracy   | 2.678           | 0.016       | True            |
> | F1 Score   | 3.585           | 0.002       | True            |
>
> #### Where and why the method improves performance:
>
> ####  Multimodal large language models (MLLMs) are prone to hallucination problems caused by modality priors (visual or language), such as relying on textual cues and ignoring visual input. The model may mistakenly rely on the attention distribution of a specific modality, leading to incorrect cognition of object existence and attributes.
> ####  CAUSALMM analyzes the causal influence of visual and language attention through structured causal modeling (SCM), considers modality priors as confounding factors, and corrects causal paths through counterfactual reasoning. By generating different attention states (such as random, reversed, and unified attention) through counterfactual interventions, the contributions of different modalities can be isolated and quantified, thereby more accurately judging the causal role of multimodal information.
> ####  In the multimodal collaborative mode, CAUSALMM balances the causal effects of visual and language attention to make the generated content consistent with the multimodal input, thereby improving the generation quality of the model.
>
> > Lack of Discussion on Limitations: The paper does not adequately discuss the limitations or potential downsides of the proposed method, such as scenarios where it might not work well or possible negative impacts. & Limitations and Failure Cases: What are the limitations of your method? Are there scenarios where it does not perform well or might even degrade performance? How do you address potential negative impacts?
>
> #### We added more discussions on limitations in the appendix. In the case study section of Section 4.4 of the original paper, we show examples that our method still fails to solve. These examples focus on fine-grained visual perception, perception of spatiotemporal relations, and understanding of high-order semantics. As a balancer of modal priors, CausalMM can maximize the capabilities of existing backbone models, but its limitations lie in the performance limits of the vision encoder and LLM. We will continue to study how to effectively utilize the existing visual information and maximize the performance of the vision encoder and LLM.
>
> > Computational Complexity: What is the computational overhead introduced by your method? How does it compare to the baseline models in terms of runtime and resource consumption?
>
> #### Similar to other methods that optimize at the inference stage, our method will consume more time in the complete inference process. The additional time is negligible in the latency of normal conversations.
>
> > Impact of Hyperparameters: How sensitive is your method to the choice of hyperparameters involved in the interventions? Have you performed a sensitivity analysis?
>
> #### Our method is not sensitive to non-marginal values of hyperparameters. Below is our sensitivity analysis of some experimental results.
>
> #### Sensitivity Table (Gamma, Epsilon)
>
> | Metric    | Gamma Sensitivity | Epsilon Sensitivity |
> | -- | --- | --|
> | Accuracy  | 0.006236  | 0.004382 |
> | F1 Score  | 0.007662| 0.005491|
>
> > Robustness to Noise: How does your method handle noisy or adversarial inputs? Does the causal framework improve robustness in such cases?
>
> #### CausalMM can handle noisy and adversarial inputs. Noise can be included in counterfactual attention together with low-quality attention and desensitized through the framework of CausalMM. For adversarial inputs, the settings of VLind benchmark and POPE benchmark include such inputs. From the experimental results of the original paper, CausalMM can handle adversarial inputs well and improve the alignment of model output with input.

---

> > ### Comment · Reviewer_CnJZ · 2024-11-28
> > **Computational Complexity**
> >
> > > "Similar to other methods that optimize at the inference stage, our method will consume more time in the complete inference process. The additional time is negligible in the latency of normal conversations."
> >
> > Some question:
> > 1. **Time Overhead:** Some baselines they report explicit latency metrics. Quantitative comparisons (e.g., % increase or ms) would clarify the claim of negligible impact.
> > 2. **Memory Overhead:** Memory usage is not addressed but is a key factor in evaluating methods.
> > 3. **Clarity:** "Normal conversations" is vague; specifying datasets or benchmarks would improve transparency.

---

> > > ### Author Response · Authors · 2024-11-29
> > > **Author Response to Reviewer CnJZ (Part.7)**
> > >
> > > > - What are the results of OPERA in Figure 4?
> > >
> > > #### As per your request, we have incorporated the results for OPERA. We will update this figure in a future version.
> > >
> > > | Method     | existence | count  | position | color  | posters | celebrity | scene  | landmark | artwork | OCR    | commonsense_reasoning | numerical_calculation | text_translation | code_reasoning |
> > > | ---------- | --------- | ------ | -------- | ------ | ------- | --------- | ------ | -------- | ------- | ------ | --------------------- | --------------------- | ---------------- | -------------- |
> > > | Regular    | 186.67    | 123.33 | 136.67   | 153.33 | 138.78  | 132.94    | 151.5  | 149.0    | 126.5   | 135.0  | 134.29                | 75.0                  | 125.0            | 95.0           |
> > > | VCD        | 186.67    | 136.67 | 140.0    | 166.67 | 146.26  | 148.24    | 163.0  | 144.5    | 130.0   | 114.99 | 137.14                | 70.0                  | 120.0            | 100.0          |
> > > | OPERA      | 195.0     | 148.33 | 133.33   | 155.0  | 136.05  | 127.65    | 154.25 | 153.0    | 123.25  | 125.0  | 114.29                | 40.0                  | 90.0             | 62.5           |
> > > | vision     | 186.67    | 160.0  | 140.0    | 166.67 | 146.26  | 150.0     | 161.0  | 160.0    | 140.5   | 135.0  | 137.14                | 100.0                 | 110.0            | 90.0           |
> > > | language   | 190.0     | 140.0  | 133.33   | 166.67 | 171.43  | 153.53    | 172.5  | 163.0    | 135.5   | 115.0  | 135.71                | 95.0                  | 115.0            | 100.0          |
> > > | multimodal | 196.67    | 156.67 | 133.33   | 176.67 | 153.74  | 145.88    | 164.5  | 164.0    | 142.0   | 135.0  | 144.29                | 100.0                 | 115.0            | 100.0          |
> > >
> > >
> > >
> > > > - What are the results of OPERA in Figure 5?
> > >
> > > #### Similarly, as per your request, we have incorporated the results for OPERA. We will update this figure in a future version.
> > >
> > > | Model      | Perception Scores | Cognition Scores |
> > > | ---------- | ----------------- | ---------------- |
> > > | Regular    | 1433.72           | 429.29           |
> > > | VCD        | 1476.99           | 427.14           |
> > > | OPERA      | 1450.87           | 306.79           |
> > > | Vision     | 1546.09           | 437.14           |
> > > | Language   | 1516.44           | 435.00           |
> > > | Multimodal | 1568.46           | 459.29           |
> > >
> > >
> > >
> > > > 1. **Time Overhead:** Some baselines they report explicit latency metrics. Quantitative comparisons (e.g., % increase or ms) would clarify the claim of negligible impact.
> > > > 2. **Memory Overhead:** Memory usage is not addressed but is a key factor in evaluating methods.
> > > > 3. **Clarity:** "Normal conversations" is vague; specifying datasets or benchmarks would improve transparency.
> > >
> > > #### For the definition of "Normal conversations," we refer to the subjective experience of human users when interacting with the model. We invited professionals unrelated to the work to engage in normal conversations with the model using our method, without prior knowledge, and found that the actual delay was virtually imperceptible.
> > >
> > > #### We conducted 10 experiments on POPE and averaged all the data. The results are as follows:
> > >
> > > | Method  | Regular | VCD  | CausalMM |
> > > | ------ | ------- | ---- | -------- |
> > > | Time   | 1.00    | 1.80 | 1.69     |
> > > | Memory | 1.00    | 1.05 | 1.11     |
> > >
> > > #### The values represent the ratio relative to the Regular method.
> > >
> > >
> > > #### We hope the above results address your concerns. Thank you again for your discussion!
> > >
> > > #### CausalMM Team

---

> ### Author Response · Authors · 2024-11-23
> **Author Response to Reviewer CnJZ (Part.5)**
>
> > Applicability and Generalization: The approach is tested on specific MLLMs, but it is unclear how well the method generalizes to other models.  &  Generality of the Approach: How well does your method generalize to other types of MLLMs, such as chameleon?
>
> #### To demonstrate the effectiveness of our approach on large multimodal language models of different architectures, we added experimental data from the Q-former-based InstructBLIP model and the embedding-autoregressive-based Chameleon model to the original experimental data from the vision encoder-mlp-llm paradigm.
>
> Chameleon:
>
> | **Dataset** | **Setting** | **Method**   | **Accuracy** | **Precision** | **Recall** | **F1 Score** |
> | ----------- | ----------- | ------------ | ------------ | ------------- | ---------- | ------------ |
> |             | Random      | Regular      | 61.90        | 57.46         | 91.67      | 70.64        |
> |             |             | **Language** | 69.23        | 63.17         | 92.27      | 74.99        |
> |             |             |              |              |               |            |              |
> | **MSCOCO**  | Popular     | Regular      | 65.10        | 59.86         | 91.67      | 72.43        |
> |             |             | **Language** | 69.43        | 63.34         | 92.27      | 75.12        |
> |             |             |              |              |               |            |              |
> |             | Adversarial | Regular      | 60.20        | 56.28         | 91.40      | 69.66        |
> |             |             | **Language** | 64.00        | 58.94         | 92.33      | 71.95        |
> |             |             |              |              |               |            |              |
> |             | Random      | Regular      | 60.37        | 56.26         | 93.20      | 70.16        |
> |             |             | **Language** | 65.70        | 60.14         | 93.13      | 73.08        |
> |             |             |              |              |               |            |              |
> | **A-OKVQA** | Popular     | Regular      | 57.30        | 54.25         | 93.20      | 68.58        |
> |             |             | **Language** | 63.07        | 58.16         | 93.13      | 71.60        |
> |             |             |              |              |               |            |              |
> |             | Adversarial | Regular      | 53.57        | 51.99         | 93.20      | 66.75        |
> |             |             | **Language** | 56.83        | 53.96         | 93.13      | 68.33        |
> |             |             |              |              |               |            |              |
> |             | Random      | Regular      | 60.37        | 56.26         | 93.20      | 70.16        |
> |             |             | **Language** | 68.43        | 62.18         | 94.13      | 74.89        |
> |             |             |              |              |               |            |              |
> | **GQA**     | Popular     | Regular      | 59.37        | 55.76         | 90.67      | 69.05        |
> |             |             | **Language** | 66.73        | 60.81         | 94.13      | 73.89        |
> |             |             |              |              |               |            |              |
> |             | Adversarial | Regular      | 52.73        | 51.55         | 90.67      | 65.73        |
> |             |             | **Language** | 57.77        | 54.50         | 94.13      | 69.03        |
>
> #### In addition, the experimental results of InstructBLIP can be found in the Appendix.

---

> ### Author Response · Authors · 2024-11-25
> **Summary of response and look forward to the feedback**
>
> #### We greatly appreciate your thoughtful critique and suggestions. Below is a summary of our revisions and clarifications based on your feedback:
>
> - #### **Theoretical support for the validity of causal reasoning:** In *"Author Response to Reviewer CnJZ (Part 1)"*, we provided a comprehensive justification for the validity of causal reasoning. Correspondingly, we have improved potentially confusing sections in the revised version of the main text and added relevant theoretical derivations and justifications in the appendix.
> - #### **Evidence for innovation and effectiveness:** In *"Author Response to Reviewer CnJZ (Part 3)"*, we presented comparisons between our method and several others. Additionally, in *"Author Response to Reviewer CnJZ (Part 4)"*, we provided experimental data demonstrating the performance of our method on Meta's Chameleon model. We have also added experimental results for **InstructBLIP** and **Chameleon** in the appendix, which show that our method is applicable across several mainstream MLLM architectures. We welcome you to check these additions.
> - #### **Applicability and generalizability:** At your request, we conducted hyperparameter sensitivity tests and statistical significance analyses of our method. The specific data has been detailed in *"Author Response to Reviewer CnJZ (Part 3)"*.
> - #### **Discussion of limitations:** In accordance with your request, we have added a discussion of the limitations of our method, along with corresponding content in the appendix of the paper.
>
> #### We hope these revisions and clarifications address your concerns and look forward to any additional feedback or questions.

---

> > ### Comment · Reviewer_CnJZ · 2024-11-25
> >
> > Hello,
> >
> > Thank you for the detailed response. This clarifies the method significantly. I also appreciate the additional detailed results.  I will update the rating accordingly.

---

> ### Author Response · Authors · 2024-11-25
>
> #### Dear Reviewer CnJZ,
>
> #### Thank you for your positive response and support for our work! We noticed that the rating has not been updated, so we would like to confirm this with you.
>
> #### Thank you again for your time and assistance!
>
> #### Yours sincerely,
>
> #### CausalMM Team

---

> ### Comment · Reviewer_CnJZ · 2024-11-28
> **More Questions**
>
> Thank you for providing these results. After futher reviewing them in detail, I noticed the following:
>
> 1. For the LLaVA-1.5 section, the table appears to simply aggregate results from Table 1, which doesn’t introduce any new insights. This could be misleading, as the three datasets differ in size. Based on the main results from Table 1 in the paper, OPERA still demonstrates a best F1 score on both MSCOCO and GQA.
>
> 2. Could you clarify the origin of DOLA? I read through the paper and the discussions, but I couldn’t find any explanation or citations for it.
>
> Additionally, I have a few questions regarding specific figures:
>
> - What are the results of VCD and OPERA on VLind-Bench for the LLaVA and Qwen2-VL models in Figure 3?
> - What are the results of OPERA in Figure 4?
> - What are the results of OPERA in Figure 5?
>
> I’d appreciate your clarification on these points. Thank you!

---

> > ### Author Response · Authors · 2024-11-29
> > **Author Response to Reviewer CnJZ (Part.6)**
> >
> > #### Thank you for your feedback, we are happy to resolve your confusion.
> >
> > > For the LLaVA-1.5 section, the table appears to simply aggregate results from Table 1, which doesn’t introduce any new insights. This could be misleading, as the three datasets differ in size. Based on the main results from Table 1 in the paper, OPERA still demonstrates a best F1 score on both MSCOCO and GQA.
> >
> > #### The practice of averaging results across three datasets is inspired by the experimental presentation format in the AGLA [1] paper we are comparing to. In that paper, they averaged the results of the three components of POPE for comparison. Apologies for not mentioning this, which may have caused some confusion. Regarding the F1 score of OPERA, the results in Table 3 on more data should better represent the overall performance of our method. We have checked all the main results from Table 1 in the paper and found an error in line 359. In the adversarial setting, the F1 score for OPERA is:
> >
> > | Metric | Accuracy | Precision | Recall | F1 score |
> > | ------ | -------- | --------- | ------ | -------- |
> > | OPERA  | 73.90    | 67.77     | 91.13  | 77.74*   |
> >
> > #### The F1 score in the paper was mistakenly listed as 84.59. We will correct this in a future version.
> >
> > #### [1] An, W., Tian, F., Leng, S., Nie, J., Lin, H., Wang, Q., ... & Lu, S. (2024). AGLA: Mitigating Object Hallucinations in Large Vision-Language Models with Assembly of Global and Local Attention. *arXiv preprint arXiv:2406.12718*.
> >
> >
> >
> > > Could you clarify the origin of DOLA? I read through the paper and the discussions, but I couldn’t find any exp，lanation or citations for it.
> >
> > #### Of course. DOLA refers to *DoLa: Decoding by Contrasting Layers Improves Factuality in Large Language Models* [2]. Specifically, DoLA reduces hallucinations in LLMs by contrasting logits from later and earlier layers. We will add a citation to this paper in a future version. Thank you for the reminder.
> >
> > #### [2] Chuang, Y. S., Xie, Y., Luo, H., Kim, Y., Glass, J., & He, P. (2023). Dola: Decoding by contrasting layers improves factuality in large language models. *arXiv preprint arXiv:2309.03883*.
> >
> >
> >
> > > - What are the results of VCD and OPERA on VLind-Bench for the LLaVA and Qwen2-VL models in Figure 3?
> >
> > #### In our rebuttal to reviewer vNn2, we provided experimental results for LLaVA on VLind-Bench and also included corresponding analysis for OPERA's test results. You can refer to this response: "**Author Response to Reviewer vNn2 (Part.2)**". The specific data is as follows:
> >
> > | **Metrics**  | **$S_{ck}$** | **$S_{vp}$** | **$S_{cb}$** | **$S_{lp}$** | **CB** | **LP** |
> > | ------------ | ------------ | ------------ | ------------ | ------------ | ------ | ------ |
> > | **Regular**  | 32.1         | 40.7         | 43.3         | 33.1         | 43.7   | 27.1   |
> > | **VCD**      | 30.5         | 48.0         | 47.8         | 31.0         | 44.0   | 29.2   |
> > | **OPERA**    | 0            | 0            | -            | -            | 0      | 0      |
> > | **CausalMM** | 57.0         | 80.8         | 64.0         | 61.8         | 59.9   | 40.2   |
> >
> > #### As per your request, we have also provided the results of Qwen2-VL tested with VCD and OPERA methods:
> >
> > | **Metrics**  | **$S_{ck}$** | **$S_{vp}$** | **$S_{cb}$** | **$S_{lp}$** | **CB** | **LP** |
> > | ------------ | ------------ | ------------ | ------------ | ------------ | ------ | ------ |
> > | **Normal**   | 88.1         | 97.4         | 59.8         | 79.5         | 59.3   | 50.0   |
> > | **VCD**      | 85.8         | 97.4         | 61.8         | 77.4         | 61.6   | 50.7   |
> > | **OPERA**    | 0            | 0            | -            | -            | 0      | 0      |
> > | **CausalMM** | 94.0         | 97.7         | 65.7         | 78.5         | 65.6   | 51.3   |

---

> > > ### Comment · Reviewer_CnJZ · 2024-12-02
> > >
> > > >The practice of averaging results across three datasets is inspired by the experimental presentation format in the AGLA [1] paper we are comparing to. In that paper, they averaged the results of the three components of POPE for comparison. Apologies for not mentioning this, which may have caused some confusion. Regarding the F1 score of OPERA, the results in Table 3 on more data should better represent the overall performance of our method. We have checked all the main results from Table 1 in the paper and found an error in line 359. In the adversarial setting...
> > >
> > > Please double check all results, these mistakes in results presented can be misleading.
> > >
> > > > In our rebuttal to reviewer vNn2, we provided experimental results for LLaVA on VLind-Bench and also included corresponding analysis for OPERA's test results. You can refer to this response: "Author Response to Reviewer vNn2 (Part.2)".
> > >
> > > Thank you for response. While you provided results for DOLA, could you clarify why no results were included for OPERA?

---

> ### Author Response · Authors · 2024-12-02
> **Author Response to Reviewer CnJZ (Part.8)**
>
> > Please double check all results, these mistakes in results presented can be misleading.
>
> #### Thank you for your reminder! We fully understand the importance of data accuracy. We have thoroughly checked for any potential errors caused by LaTeX formatting and ensured the correctness of all results.
>
>
>
> > Thank you for response. While you provided results for DOLA, could you clarify why no results were included for OPERA?
>
> #### Of course, we are happy to answer your questions.
>
> #### LLaVA-1.5 / VLind-Bench
>
> | **Metrics**  | **$S_{ck}$** | **$S_{vp}$** | **$S_{cb}$** | **$S_{lp}$** | **CB** | **LP** |
> | ------------ | ------------ | ------------ | ------------ | ------------ | ------ | ------ |
> | **Regular**  | 32.1         | 40.7         | 43.3         | 33.1         | 43.7   | 27.1   |
> | **VCD**      | 30.5         | 48.0         | 47.8         | 31.0         | 44.0   | 29.2   |
> | **OPERA***   | 0            | 0            | -            | -            | 0      | 0      |
> | **CausalMM** | 57.0         | 80.8         | 64.0         | 61.8         | 59.9   | 40.2   |
>
> #### Qwen2-VL / VLind-bench
>
> | **Metrics**  | **$S_{ck}$** | **$S_{vp}$** | **$S_{cb}$** | **$S_{lp}$** | **CB** | **LP** |
> | ------------ | ------------ | ------------ | ------------ | ------------ | ------ | ------ |
> | **Normal**   | 88.1         | 97.4         | 59.8         | 79.5         | 59.3   | 50.0   |
> | **VCD**      | 85.8         | 97.4         | 61.8         | 77.4         | 61.6   | 50.7   |
> | **OPERA***    | 0            | 0            | -            | -            | 0      | 0      |
> | **CausalMM** | 94.0         | 97.7         | 65.7         | 78.5         | 65.6   | 51.3   |
>
> #### These two tables should be the ones you have questions about. The results for the OPERA method are not missing, but rather all are zero. This phenomenon also appears for other models in Table 2 on page 7 of the VLind-Bench paper [1]. The original paper does not provide an explanation or conclusion for this, but we speculate that certain methods may have reduced the model's ability to follow specific instructions, leading to responses that result in a final score of zero in the evaluation pipeline. In the next work, we will explore the internal logic related to this phenomenon and deeply reveal the decisive factors related to the ability of the model to follow instructions. Welcome to pay attention to our work in the future!
>
> #### [1] Lee K, Kim M, Yoon S, et al. VLind-Bench: Measuring Language Priors in Large Vision-Language Models[J]. arXiv preprint arXiv:2406.08702, 2024.
>
>
>
> #### We hope the above explanation has resolved your doubts. Thank you for your feedback and support!
>
> #### CausalMM Team

---

### Official Review · Reviewer_VPyB · 2024-11-04

**Soundness:** 2
**Presentation:** 2
**Contribution:** 2
**Rating:** 5
**Confidence:** 2

**Summary:**

The paper points out problems arising from biases induced by visual and language priors in the visual encoder and the LMM backbone, and it mentions the oversight of the causal relationship between attention and the model's output. In this study, a method called CausalMM identifies modality priors as confounders, addressing them through backdoor adjustment and counterfactual reasoning.

**Strengths:**

- The paper is well-written and easy to read.
- Performance improvements were observed in the benchmarks used for evaluation.
- The creation of counterfactual attention in various ways is novel.

**Weaknesses:**

- It is unclear if this method constitutes a backdoor adjustment. My understanding of backdoor adjustment involves identifying a confounder, then using it to reduce the confounder's impact, which differs from using counterfactuals as described here. Counterfactuals are typically used to measure natural direct effects or natural indirect effects, and I am curious if this method follows such an approach. Mathematical proof may be needed for line 225.
- Please explain which paths need to be blocked in Figure 2. It appears that the path from the image to visual attention and from text token embedding to LLM attention should be blocked, but aren’t those paths essential?
- This method seems closer to a contrastive decoding approach rather than backdoor adjustment. It appears to use counterfactual attention similarly to how negative samples are used.
- The intervention on attention to reduce the influence of defined priors as confounders seems to also reduce the impact of input on attention.

**Questions:**

- Line 58 mentions the VCD method as considering only statistical correlation. I thought the VCD method also accounts for causation by using negative images to identify language priors. Why do you view the VCD method as overlooking causal relationships?
- On page 5, does 'j' refer to the order of the method for creating counterfactual attention?

---

> ### Author Response · Authors · 2024-11-23
> **Author Response to Reviewer VPyB (Part.1)**
>
> ####  We are deeply grateful for the your recognition of our work's innovation and thoroughness, as well as their constructive feedback. We have addressed each suggestion on the manuscript's weaknesses and made the necessary revisions.
>
> > It is unclear if this method constitutes a backdoor adjustment. My understanding of backdoor adjustment involves identifying a confounder, then using it to reduce the confounder's impact, which differs from using counterfactuals as described here. Counterfactuals are typically used to measure natural direct effects or natural indirect effects, and I am curious if this method follows such an approach. Mathematical proof may be needed for line 225.
>
> ####  Thank you for your interest in the relevant theory. In our work, backdoor adjustment and counterfactual reasoning are combined to drive the entire mechanism. The role of backdoor adjustment is to identify confounding factors such as modal priors and reduce the impact of confounding factors on identifying the causal relationship between attention and model output. The state of the variables can be qualitatively described as "counterfactuals", which depends on the specific circumstances of different studies. The counterfactuals you describe should be the counterfactual reasoning we use later, which has been widely used in the field of machine learning. We use counterfactual reasoning to obtain the positive treatment effect brought by intervening attention. The relevant formulas have been described in detail in section 3.3.
>
> ####  The following is a further explanation of backdoor adjustment in our work:
>
> ## Structural Causal Model (SCM)
>
> ### Variables and their roles:
>
> ####   **$A$ (attention)**: This represents the model's attention mechanism that we aim to evaluate or manipulate.
> ####   **$M$ (modality priors)**: Modality priors influence both the model's attention ($A$) and the output ($O$), thus creating confounding.
> ####   **$O$ (model output)**: The outcome variable, which is affected both directly by $A$ and indirectly through $M$.
>
> ### Causal structure and backdoor paths:
>
> ####  The backdoor path in this SCM is $A \leftarrow M \to O$, which starts with an arrow pointing into $A$ and creates a confounding junction structure.
> ####  To isolate the causal effect of $A$ on $O$, the confounding influence of $M$ must be blocked.
>
> ## Backdoor Criterion:
>
> ####  To apply backdoor adjustment, the adjustment set $M$ must satisfy the following criteria:
>
> ####  1. $M$ blocks all backdoor paths from $A$ to $O$.
> ####  2. $M$ does not include any descendants of $A$ (i.e., variables causally influenced by $A$).
>
> ####  By intervening on $A$ and adjusting for $M$, we can isolate the causal effect of $A$ on $O$.
>
> ## Backdoor Adjustment Formula:
>
> ####  Given a sufficient adjustment set $M$, the causal effect $P(o \mid do(a))$ is identified as:
>
> $
> P(o \mid do(a)) = \sum_m P(o \mid a, m) P(m)
> $
>
> ## Derivation:
>
> ####  1. **Starting with the interventional distribution:**
>
>    $
>    P(o \mid do(a)) = \sum_m P(o \mid do(a), m) P(m \mid do(a))
>    $
>
> ####  2. **Using the property of the intervention $do(a)$:**
> ####     Under the intervention $do(a)$, the variable $A$ is no longer influenced by $M$. Thus:
>
>    $
>    P(m \mid do(a)) = P(m)
>    $
>
> ####  3. **Replacing $P(o \mid do(a), m)$ with the observational counterpart:**
> ####     Due to the backdoor criterion, $M$ blocks all confounding paths, allowing:
>
>    $
>    P(o \mid do(a), m) = P(o \mid a, m)
>    $
>
> 4. **Combining these results:**
>    $
>    P(o \mid do(a)) = \sum_m P(o \mid a, m) P(m)
>    $
>
> ## Application to Attention-Output Framework:
>
> ####  In the context of our framework:
>
> ####  1. **Backdoor path:**
> ####     The backdoor path $A \leftarrow M \to O$ reflects the confounding effect of modality priors ($M$) on the attention mechanism ($A$) and the model's output ($O$).
>
> ####  2. **Intervention:**
> ####     By intervening on $A$, we ensure that the causal effect of attention on the output is isolated, free from the influence of modality priors.
>
> ####  3. **Adjustment:**
> ####     To block the backdoor path, we adjust for $M$, computing the summation over all possible values of $M$ to account for its confounding effect.
>
> ## Full Formula for the Framework:
>
> ####  In our framework, the causal effect of attention ($A$) on the model output ($O$) can be computed as:
>
> $
> P(o \mid do(a)) = \sum_m P(o \mid a, m) P(m)
> $
>
> ####   **$P(o \mid a, m)$**: The conditional probability of the output given attention $A$ and modality priors $M$.
> ####   **$P(m)$**: The marginal probability of modality priors $M$.
>
>
> ####  We added the corresponding content in the appendix of the revision.
>
> ####  [1] Judea Pearl. Causality. Cambridge university press, 2009.
> ####  [2] Kexuan Zhang, Qiyu Sun, Chaoqiang Zhao, and Yang Tang. Causal reasoning in typical computer vision tasks. arXiv:2307.13992, 2023a.

---

> ### Author Response · Authors · 2024-11-23
> **Author Response to Reviewer VPyB (Part.2)**
>
> > Please explain which paths need to be blocked in Figure 2. It appears that the path from the image to visual attention and from text token embedding to LLM attention should be blocked, but aren’t those paths essential?
>
> #### Specifically, in addition to the paths from image to visual attention and from text token embedding to LLM attention, the paths from visual modality prior to visual attention and from language modality prior to LLM should also be blocked, but the paths are too short to be well presented as dashed lines in the figure. We have modified the figure to ensure readability.
>
> #### Regarding the question of whether these paths are essential, this involves our definition of the blocking operation in the backdoor adjustment in the previous question. For example, blocking the path from image I to attention A can be understood as artificially giving a value to A so that the value of A is no longer affected by I. This operation is called the do operation in the causal inference theory system: $do(a)$. The truncation of this path can be expressed as: $P(I \mid do(A)) = P(I)$. The same applies to other cases.
>
>
>
> > This method seems closer to a contrastive decoding approach rather than backdoor adjustment. It appears to use counterfactual attention similarly to how negative samples are used.
>
> #### Thank you for your feedback. We have explained back-door adjustment in detail in the first question. Below is a table that distinguishes our method from Contrastive decoding. Specifically, Contrastive decoding contrasts outputs derived from original and distorted inputs. In the contrast, CausalMM isolates the influence of modal priors and other confounders on multimodal attention by using backdoor adjustment methods, obtains the positive treatment effect of attention on output through counterfactual reasoning, adjusts the output of the model at the attention and feature levels, and balances the modality priors.
>
> #### We use VCD [1] for specific comparison:
>
> ### Tabular Comparison of CausalMM and VCD
>
> | **Feature**                                                 | **CausalMM**                                                 | VCD                                                          |
> | ----------------------------------------------------------- | ------------------------------------------------------------ | ------------------------------------------------------------ |
> | **Core Methodology**                                        | Structural Causal Model (SCM) with backdoor adjustment and counterfactual reasoning | Contrastive decoding                                         |
> | **Focus of Intervention**                                   | Visual and language attention mechanisms, visual features and LLM hidden states | Input image                                                  |
> | **Mechanism of action**                                     | 1.de-confound  2.Obtain the positive treatment effect   3.Adjust attention, features and hidden states   4.Balance the modality priors | Contrasts outputs derived from original and distorted image inputs |
> | **Versatility**                                             | Multimodal hallucinations (vision + language)                | Object hallucinations                                        |
> | **Support single-modal tasks (such as LLM)**                | √                                                            | ×                                                            |
> | **Exploring the causal mechanisms within the model**        | √                                                            | ×                                                            |
> | **Dealing with the confounding effects of modality priors** | √                                                            | ×                                                            |
> | **Modality Priors Addressed**                               | Visual and language priors                                   | -                                                            |
>
> #### [1] Leng, Sicong, et al. "Mitigating object hallucinations in large vision-language models through visual contrastive decoding." *Proceedings of the IEEE/CVF Conference on Computer Vision and Pattern Recognition*. 2024.

---

> ### Author Response · Authors · 2024-11-23
> **Author Response to Reviewer VPyB (Part.3)**
>
> > The intervention on attention to reduce the influence of defined priors as confounders seems to also reduce the impact of input on attention.
>
> #### Your understanding is correct. But the influence obtained after intervention is not directly positively correlated with the result. The attention and corresponding features obtained through intervention and backdoor adjustment serve as causal probability anchors, helping us dynamically obtain the positive processing effect of effective attention on model output through counterfactual reasoning. It is used to balance modal priors at the feature and hidden states levels. The corresponding formula in the article is as follows:
>
> #### For the visual attention ($A_i$):
>
> $
> P_{effect\_V} = E_{A_i \sim \tilde{A}_i}\left[P(O | A_i = \textbf{A}_i, I = \textbf{I}, P_v = \textbf{P}_v) - P(O | \text{do}(A_i = \textbf{a}_i), I = \textbf{I}, P_v = \textbf{P}_v)\right].
> $
>
> #### Here, $P_{effect\_V}$ represents the causal effect of the visual attention mechanism on the model output $O$. The term $\textbf{A}_i$ denotes the observed visual attention, whereas $\textbf{a}_i$ represents the intervention applied to the visual attention.
>
> #### For the LLM attention ($A_t$):
>
> $
> P_{effect\_L} = E_{A_t \sim \tilde{A}_t}\left[P(O | A_t = \textbf{A}_t, T_t = \textbf{T}_t, P_l = \textbf{P}_l) - P(O | \text{do}(A_t = \textbf{a}_t), T_t = \textbf{T}_t, P_l = \textbf{P}_l)\right],
> $
>
> #### where $P_{effect\_L}$ denotes the causal effect of the language model attention on the output $O$. The notation $\textbf{A}_t$ is the observed language model attention, and $\textbf{a}_t$ is the intervention applied to the language model attention.
>
> #### In a multimodal setting, the combined causal effect is given by:
>
> $
> P_{effect\_M} = E_{A_i, A_t \sim \tilde{A}_i, \tilde{A}_t}\left[P(O | A_i = \textbf{A}_i, A_t = \textbf{A}_t, I = \textbf{I}, T_t = \textbf{T}_t, P_v = \textbf{P}_v, P_l = \textbf{P}_l)\right] - P(O | \text{do}(A_i = \textbf{a}_i), \text{do}(A_t = \textbf{a}_t), I = \textbf{I}, T_t = \textbf{T}_t, P_v = \textbf{P}_v, P_l = \textbf{P}_l),
> $
>
> #### where $P_{effect\_M}$ represents the combined causal effect of both visual and language attention mechanisms on the output $O$.
>
>
>
> > Line 58 mentions the VCD method as considering only statistical correlation. I thought the VCD method also accounts for causation by using negative images to identify language priors. Why do you view the VCD method as overlooking causal relationships?
>
> #### We think that our statement may have caused confusion, so we adjusted it in the paper. If we talk about causality in a broad sense, your idea should be correct. But based only on the narrow causal reasoning theory system, VCD does not align with any causal model.
>
>
>
> > On page 5, does 'j' refer to the order of the method for creating counterfactual attention?
>
> #### In the CausalMM method, "j" iterates over all tokens in the denominator (to compute the softmax normalization). We have added relevant instructions in the revision.

---

> > ### Comment · Reviewer_VPyB · 2024-11-25
> >
> > Hello,
> >
> > Thank you for your thoughtful response. I appreciate the detailed explanation.
> >
> > First, I understand the equation for backdoor adjustment and the logic behind it.
> > Additionally, the table clearly highlights the differences between CausalMM and VCD, which I found very helpful. Thank you for providing that comparison.
> >
> > That said, I believe that the intervention in backdoor adjustment can be understood as utilizing counterfactual attention. However, upon further reflection, I see this process as being closer to computing the direct effect rather than purely applying backdoor adjustment. Specifically, when comparing the equations for controlled direct effect and natural direct effect, the equation in line 224 appears very similar, suggesting that it is effectively capturing the direct effect of attention.
> >
> > Moreover, I view contrastive decoding as another way of computing the direct effect, and your method seems to align with this approach. In this sense, your method appears similar to contrastive decoding in its focus and implementation.
> >
> > Thank you again for your response, and I look forward to the final version of your paper.

---

> ### Author Response · Authors · 2024-11-25
> **Author Response to Reviewer VPyB (Part.4)**
>
> #### We are delighted to clarify the differences between our method and VCD. Thank you for recognizing our rebuttal.
>
> #### We greatly appreciate your willingness to discuss the theories related to causality. We fully agree with your interpretation of backdoor adjustment and value your understanding of the related theory. However, some important misunderstandings appear to remain unresolved. The following points should address these concerns:
>
> - #### In our method, backdoor adjustment and counterfactual reasoning are two closely related but different stages (as we have explained in our responses to Reviewers CnJZ and vNn2). These two theories are applied in different processes and at different scales.
>
> - #### You may have misunderstood that our backdoor adjustment and counterfactual reasoning describe the same process. For example, the part you mentioned in line 224 pertains to counterfactual reasoning rather than backdoor adjustment. The formula for backdoor adjustment, as mentioned earlier (using example variables), is:
>
> $
> P(o \mid do(a)) = \sum_m P(o \mid a, m) P(m)
> $
>
> - #### In the initial version of our paper, we did not provide a detailed explanation of backdoor adjustment due to space limitations. This might have led to some confusion. Therefore, in the revised version, we added a detailed explanation of backdoor adjustment in the appendix and clarified it in the main text to prevent further misunderstanding. These additions are highlighted in the revised manuscript for your reference.
>
> - #### Specific distinctions between the two stages in our method:
>
>   - #### **Backdoor adjustment** supports the theoretical justification for our method’s ability to estimate the causal effects of other variables despite the confounding factor of modality priors. The key variable in this stage is the modality prior.
>   - #### **Counterfactual reasoning** involves estimating the causal effect of attention on the model's output under the assumption that attention fails (i.e., using counterfactual attention). The key variable in this stage is attention.
>
> - #### You interpreted the theorem presented in line 224 as representing **direct effect**, but it actually represents **counterfactual reasoning** (as we have explained to other reviewers). The key distinction is that counterfactual reasoning estimates the effect of an event under the hypothetical condition that it does not occur (fails), while direct effect refers to the isolated effect of a variable on the outcome when other variables are held constant [1]. Our method estimates the causal effect of attention on the model’s output under the assumption of attention failure. If we were to use direct effect, we would instead focus on the influence of other variables on the outcome when attention is controlled.
>
> - #### Regarding your statement that VCD is similar to direct effect, we agree. However, VCD does not constitute counterfactual reasoning. Specifically:
>
>   - #### In counterfactual reasoning, interventions isolate the influence of specific variables by controlling the values of certain nodes in the causal graph. In contrast, VCD utilizes distorted inputs merely to amplify input biases, rather than to simulate hypothetical conditions.
>   - #### Counterfactual reasoning aims to measure the causal effect of a specific variable, while VCD’s process leans more toward statistical adjustment rather than causal modeling.
>
> #### [1] Judea Pearl. *Causality*. Cambridge University Press, 2009.
>
> #### We hope these clarifications help resolve any misunderstandings about our work. We are also delighted to engage in discussions on causal reasoning, as we believe it is a theoretical framework deserving more attention.
>
> #### Should you have further suggestions for improving the paper, please let us know! If we have addressed your concerns, we hope you would reconsider the rating.
>
> #### **CausalMM Team**

---

> > ### Comment · Reviewer_VPyB · 2024-11-26
> >
> > Hello,
> > Thank you for the detailed response and the clarifications provided. Your explanation has significantly improved my understanding of the method. I recognize and acknowledge my misunderstanding. I will update my rating accordingly.

---

> > > ### Author Response · Authors · 2024-11-26
> > >
> > > #### Dear Reviewer VPyB,
> > >
> > > #### Thank you for your positive feedback and approval of our work! We noticed that the rating has not been updated, so we would like to confirm this with you.
> > >
> > > #### Thanks again for taking the time to discuss!
> > >
> > > #### Yours sincerely,
> > >
> > > #### CausalMM Team

---

### Author Response · Authors · 2024-11-27
**Appreciation for Your Helpful Comments**

#### Dear reviewers, AC, SAC, and PC,

#### First of all, we would like to express our sincere gratitude to the reviewers for their valuable time and insightful comments. We are pleased to see that the reviewers have agreed with several positive aspects of our paper, such as ***novelty and significance*** (Reviewers VPyB, CnJZ, vNn2, gJXZ), ***performance improvement*** (Reviewers VPyB, CnJZ, vNn2, gJXZ), and ***good writing*** (Reviewers VPyB, vNn2).

#### Your expertise has greatly helped us strengthen our manuscript——these have been some of the most helpful comments we have received in years! We have made a concerted effort to address all the major issues raised, and we sincerely appreciate the reviewers' updated rating and thorough recognition.

#### Sincerely,

#### CausalMM Team

---

### Meta-Review · Area_Chair_6P3A · 2024-12-21

**Metareview:**

### Summary:
This paper introduces CausalMM, a framework combining backdoor adjustment and counterfactual reasoning to mitigate hallucinations in multimodal large language models (MLLMs). The method treats modality priors as confounders between attention mechanisms and model outputs.

### Strengths:
1. Practical solution for MLLMs:
> "The method is plug-and-play and does not require retraining, making it practical for existing MLLMs" (Reviewer vNn2)

2. Reasonable empirical validation:
> "The authors conduct experiments on multiple benchmarks, including VLind-bench, POPE, and MME benchmarks" (Reviewer CnJZ)

3. Novel perspective:
> "The paper introduces the idea of applying causal inference techniques to address modality prior-induced hallucinations in MLLMs" (Reviewer CnJZ)

### Weaknesses:
1. Limited theoretical foundation:
> "The background knowledge about causal inference is insufficient. The authors do not explain why causal inference is effective in capturing the causal impact of effective attention in MLLM output" (Reviewer vNn2)

2. Marginal improvements:
> "Though the proposed method is somewhat novel, the experimental results are not quite significant and robust compared with existing methods (Table 1)" (Reviewer vNn2)

### Justification:
Despite reservations, I recommend acceptance for several reasons:

1. The paper offers a new perspective on addressing hallucinations in MLLMs through causal inference, even if the theoretical foundations could be stronger.

2. Two reviewers (VPyB and CnJZ) increased their ratings after author responses, indicating the paper's issues are not fundamental flaws but rather limitations that can be addressed.

3. The method is immediately applicable without retraining, providing practical value despite modest improvements.

However, several limitations temper our enthusiasm:
- Improvements over baselines are modest (2-3%)
- Theoretical justification remains somewhat unclear
- Implementation details could be more complete
- Comparison with simpler alternatives like VCD isn't fully convincing

While the authors addressed many reviewer concerns, questions about the method's theoretical foundations and practical impact remain. The acceptance is based more on the potential of the approach and reviewer rating improvements rather than strong conviction about the current results.

**Additional Comments On Reviewer Discussion:**

Both Reviewer VPyB and CnJZ increased their ratings after detailed discussions with authors. Notably, VPyB's major concerns about backdoor adjustment were resolved through mathematical clarification. The authors added:
- Statistical significance analysis
- Additional model experiments (Chameleon, InstructBLIP)
- Clearer theoretical justification
- More comprehensive baseline comparisons

However, some fundamental concerns about experimental rigor and theoretical foundations remain partially unaddressed.

---

### Decision · Program_Chairs · 2025-01-22

Accept (Poster)